# Female sexual behavior in mice is controlled by kisspeptin neurons

Vincent Hellier[1], Olivier Brock[1,2], Michael Candlish[3], Elodie Desroziers[1], Mari Aoki[3], Christian Mayer [4], Richard Piet[5], Allan Herbison [5], William Henry Colledge[6], Vincent Prévot[7], Ulrich Boehm[3] & Julie Bakker[1,2]

Sexual behavior is essential for the survival of many species. In female rodents, mate preference and copulatory behavior depend on pheromones and are synchronized with ovulation to ensure reproductive success. The neural circuits driving this orchestration in the brain have, however, remained elusive. Here, we demonstrate that neurons controlling ovulation in the mammalian brain are at the core of a branching neural circuit governing both mate preference and copulatory behavior. We show that male odors detected in the vomeronasal organ activate kisspeptin neurons in female mice. Classical kisspeptin/Kiss1R signaling subsequently triggers olfactory-driven mate preference. In contrast, copulatory behavior is elicited by kisspeptin neurons in a parallel circuit independent of Kiss1R involving nitric oxide signaling. Consistent with this, we find that kisspeptin neurons impinge onto nitric oxide-synthesizing neurons in the ventromedial hypothalamus. Our data establish kisspeptin neurons as a central regulatory hub orchestrating sexual behavior in the female mouse brain.

[1] GIGA Neurosciences, Neuroendocrinology, University of Liege, 4000 Liege, Belgium. [2] Netherlands Institute for Neuroscience, 1105 BA Amsterdam, The Netherlands. [3] Experimental Pharmacology, Center for Molecular Signaling (PZMS), Saarland University School of Medicine, 66421 Homburg, Germany. [4] NYU School of Medicine, New York, NY 10016, USA. [5] Center for Neuroendocrinology and Department of Physiology, University of Otago, Dunedin 9054, New Zealand. [6] Reproductive Physiology Group, Department of Physiology, Development, and Neuroscience, University of Cambridge, Cambridge CB2 3EG, UK. [7] Laboratory of Development and Plasticity of the Neuroendocrine Brain, Jean-Pierre Aubert Research Center, Inserm U1172, F- 59000 Lille Cedex, France. Vincent Hellier and Olivier Brock contributed equally to this work. Ulrich Boehm and Julie Bakker jointly supervised this work. Correspondence and requests for materials should be addressed to U.B. (email: ulrich.boehm@uks.eu) or to J.B. (email: jbakker@uliege.be)

Female copulatory behaviors are exquisitely orchestrated by sex hormones in order to coincide with ovulation and thus to ensure the highest possible chance of fertilization[1]. Female rodents typically control the initiation and timing of copulatory contacts with males[2]. This is achieved by a succession of precopulatory behaviors bringing the female in contact with males and is driven by sexual motivation and mate preferences. Once in direct contact with the male, the female will display receptive behaviors, such as the lordosis posture, which is necessary for intromission. The neural circuits and the individual neurons underlying the coordination of sexual motivation and mate preference with ovulation have, however, remained elusive.

Mice are nocturnal animals and heavily rely on olfactory cues, such as pheromones, to identify potential mates[3]. Pheromones are detected and processed by a highly specialized neural circuit, the accessory olfactory system, which initiates in the vomeronasal organ (VNO) in the nasal septum. The vomeronasal pathway runs in parallel to the main olfactory system originating in the main olfactory epithelium (MOE). The accessory olfactory system has been functionally linked to a range of innate behaviors including reproduction[4–8]. How exactly pheromones impinge on both reproductive physiology and behavior has only recently begun to emerge. Signals triggered by pheromones in the neuroepithelium of the VNO are transmitted to gonadotropin-releasing hormone (GnRH) neurons in the reproductive center of the neuroendocrine brain[9]. GnRH neurons are located in the preoptic area of the anterior hypothalamus and control the hypothalamus–pituitary gonadal axis by releasing the GnRH peptide from axon terminals in the median eminence[10]. GnRH then acts on gonadotrope cells in the anterior pituitary to trigger secretion of the gonadotropins; luteinizing hormone (LH) and follicle-stimulating hormone (FSH). LH and FSH in turn regulate sex hormone production (i.e., estradiol in females and testosterone in males) and the maturation of the gametes. In rodents, pheromones are known to modulate GnRH neuronal activity in a sex-dependent manner[11,12]. Exposure to female pheromones activates GnRH neurons in male mice consequently leading to a LH/testosterone surge, and vice versa male pheromones induce LH release in female mice[13].

Sex hormones provide gonadal feedback to the brain; however, GnRH neurons do not express the appropriate steroid hormone receptors and are thus not direct targets of this feedback regulation[10]. Instead, gonadal sex steroid feedback is indirectly relayed to GnRH neurons. One critical neuronal population relaying sex steroid feedback to GnRH neurons lies in the rostral periventricular area of the third ventricle (RP3V) of the hypothalamus and produces the neuropeptide kisspeptin. Kisspeptin is a potent stimulator of GnRH neuronal activity (acting via its receptor Kiss1R, also known as GPR54)[14,15] and controls ovulation by driving the LH surge[16,17]. RP3V kisspeptin neurons display a profound sexual dimorphism with greater neuron numbers in female than in male mice[18] consistent with a pivotal role of these cells in female reproduction.

We recently found that the RP3V kisspeptin neuronal population in female mice is specifically activated by male but not by female pheromones[19], raising the possibility that these neurons serve reproductive functions in addition to providing estradiol feedback to GnRH neurons. To dissect the functional role of RP3V kisspeptin neurons in reproduction, we applied a combination of complementary genetic strategies. We demonstrate that the RP3V kisspeptin neuronal population is necessary for the expression of both male-directed mate preference and lordosis behavior using specific viral-based cell ablation techniques and optogenetic stimulation. We then analyzed the neural circuitry downstream of RP3V kisspeptin neurons and found that mutant mice lacking GnRH secretion in adulthood

also failed to show any male-directed preference. These data suggest that kisspeptin neurons act through GnRH neurons to trigger olfactory-driven mate preferences in female mice. In contrast, lordosis behavior did not depend on GnRH neurons downstream of RP3V kisspeptin neurons. To dissect the downstream neural circuitry-mediating lordosis in female mice, we then employed a combination of genetic transsynaptic tracing and viral tract tracing from RP3V kisspeptin neurons. We found that a subset of neurons in the ventrolateral part of the ventromedial hypothalamus (VMHvl) that express nitric oxide synthase (nNOS) and are communicating with kisspeptin neurons. Consistent with the idea that nitric oxide (NO) is a key neurotransmitter downstream of kisspeptin neurons, female mice deficient in nNOS showed a strong decrease in lordosis behavior. Taken together, our results demonstrate that kisspeptin governs both mate preference and sexual motivation in female mice, indicating that sexual behavior and ovulation are coordinated by the same neuropeptide.

## Results

**Male odors activate RP3V kisspeptin neurons**. We previously found that RP3V kisspeptin neurons are specifically activated by male odors derived from either urine[19] or soiled bedding in female mice. To determine the olfactory input pathway impinging onto these cells, we selectively ablated the vomeronasal organ (VNO) by surgical removal and/or the main olfactory epithelium (MOE) by intranasal infusion with a zinc sulfate (ZnSO₄) solution in female *C57Bl/6J* mice (see general remarks on behavioral tests in "Methods" section and Supplementary Table 1 for details on hormonal treatments prior to testing). We then tested male odor-triggered kisspeptin neuron activation in these animals by using c-Fos as a marker. Removal of the VNO (VNOx), but not ablation of the MOE (MOEx), completely eliminated the ability of male odors contained in soiled bedding to activate RP3V kisspeptin neurons in ovariectomized female mice supplemented with estradiol and progesterone (OVX+E+P) (VNOi group vs. control group, $P < 0.001$; MOEx group vs. control group, $P = 0.001$; VNOx group vs. control group, $P > 0.99$; Dunn's multiple comparison test; Fig. 1a). These results demonstrate that pheromonal input triggers c-Fos expression in kisspeptin neurons via the vomeronasal pathway. The number of kisspeptin neurons was not affected by either VNOx or zinc sulfate treatment.

**RP3V kisspeptin neurons trigger mate preferences**. Since pheromonal cues are critical for mate recognition[20,21], we next asked whether the kisspeptin peptide plays a role in olfactory mate preference. To address this question, we analyzed mice lacking a functional *Kiss1* gene[22]. Kisspeptin knockout (*Kiss*−/−; OVX+E+P) mice failed to show any male-directed preference (one-sample *t* test (H0: mean equals 0); $P = 0.42$; Fig. 1b), whereas control littermates (OVX+E+P) displayed robust preferences for the male (one-sample *t* test (H0: mean equals 0); $P < 0.001$; Fig. 1b). A single subcutaneous (sc) injection of kisspeptin (Kp-10; 0.52 μg kg⁻¹) triggered a male-directed preference in Kiss−/− (OVX+E+P) females (one-sample *t* test (H0: mean equals 0); $P < 0.001$; Fig. 1b). These data implicate the kisspeptin neuropeptide in olfactory-driven partner preference. Kisspeptin is, however, not only expressed in the RP3V, but also in neurons located in the arcuate nucleus of the hypothalamus (ARC) in the adult rodent brain[18]. To analyze the specific role of the RP3V kisspeptin neuron population in olfactory mate preference, we ablated these cells by injecting an adeno-associated virus (AAV) encoding a Cre recombinase-dependent caspase 3[23] bilaterally into the RP3V of mice expressing Cre in kisspeptin neurons (*KissIC*). Caspase 3 kills the Cre-expressing cells by inducing apoptosis[23]. Stereotaxic

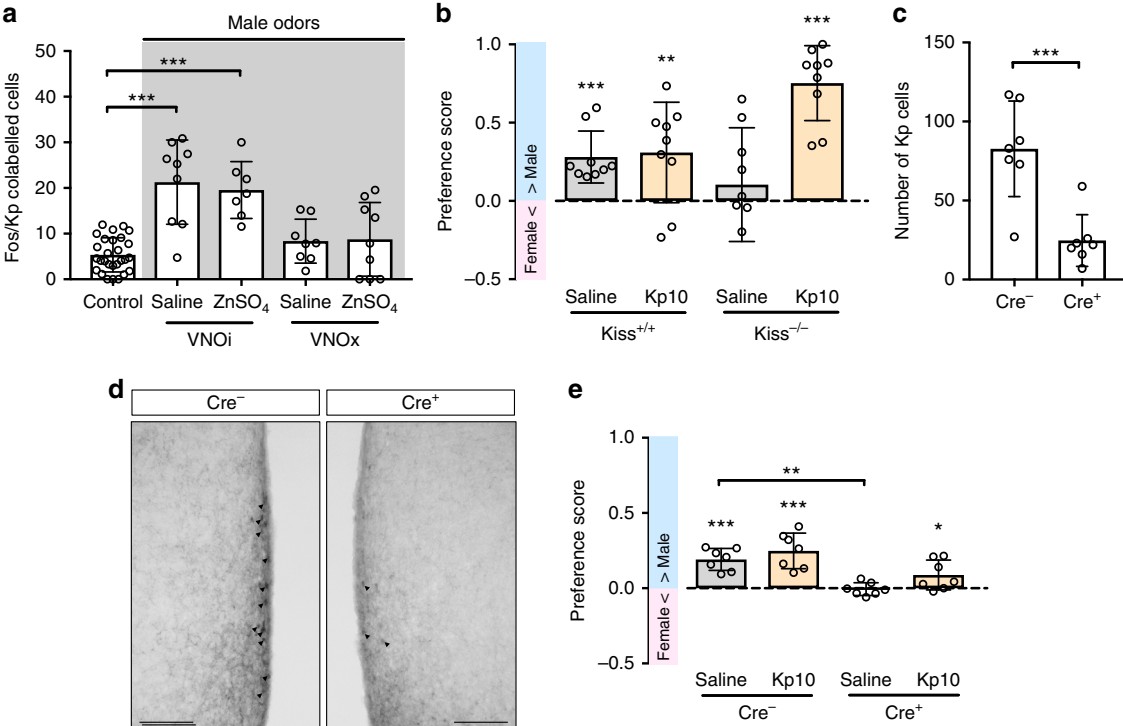

**Fig. 1** RP3V kisspeptin neurons are part of a motivational circuit triggered by male olfactory cues. **a** Removal of the vomeronasal organ (VNOx) but not ablation of the main olfactory epithelium by intranasal infusion of a zinc sulfate solution ($ZnSO_4$) with the vomeronasal organ intact (VNOi) disrupted male odor-induced activation of RP3V kisspeptin neurons as determined by the percentage of Fos/Kp co-labeled cells. ***$P \leq 0.001$; Dunn's multiple comparison test; $n = 28/9/7/8/9$. **b** Kisspeptin knockout ($Kiss^{-/-}$) female mice do not show a male-directed preference, whereas female control littermates displayed a preference for the male. Strikingly, a single peripheral injection with kisspeptin (Kp-10) at a dose of 0.52 µg kg$^{-1}$ induced a very strong preference for the male in $Kiss^{-/-}$ female mice. **$P \leq 0.01$; ***$P \leq 0.001$; one-sample $t$ test (H0: mean equals 0); $n = 9$ per group. **c** Stereotaxic injection with an AAV encoding a Cre-dependent caspase bilaterally into the RP3V led to a ~70% decrease in the number of RP3V kisspeptin (Kp) cells in $Cre^+$ compared to $Cre^-$ females. ***$P \leq 0.001$; unpaired $t$ test; $n = 7$. **d** Photomicrographs showing viral ablation of RP3V kisspeptin neurons in KissIC mice (left: $Cre^-$; right $Cre^+$). **e** Viral ablation of RP3V kisspeptin cells disrupted male-directed preferences in KissIC mice ($Cre^+$), whereas a peripheral injection with Kp-10 induced a male-directed preference. *$P \leq 0.05$; ***$P \leq 0.001$; one-sample $t$ test (H0: mean equals 0); **$P \leq 0.01$; Tukey's multiple comparison test; $n = 7$ per group. Scale bar represents 100 µm. Bars represent the mean ± SEM. For all experimental details, see supplementary Table 1

viral delivery into the RP3V led to a 71% decrease in kisspeptin-immunoreactive cells in $Cre^+$ females compared to control $Cre^-$ animals (unpaired $t$ test; $P < 0.001$; Fig. 1c, d), suggesting efficient ablation of this neuronal population. In contrast, kisspeptin immunoreactivity in the ARC was not affected in this experimental paradigm (unpaired $t$ test; $P = 0.81$, Supplementary Fig. 1). Strikingly, $Cre^+$ females failed to show any male-directed preferences after acute ablation of the RP3V kisspeptin neuron population (one-sample $t$ test (H0: mean equals 0); $P = 0.73$; Fig. 1e), whereas a single sc injection of Kp-10 was sufficient to induce a male-directed preference in these female (OVX+E+P) mice (one-sample $t$ test (H0: mean equals 0); $P = 0.041$; Fig. 1e). Taken together, these data demonstrate that RP3V kisspeptin neurons could be an essential component of the neural circuits downstream of the vomeronasal organ mediating pheromone-driven mate preference in female mice.

**RP3V kisspeptin neurons are essential for lordosis.** Pheromone-triggered mate preference ultimately leads to the display of copulatory behaviors. We therefore next investigated the role of the RP3V kisspeptin neurons in lordosis behavior, which is characterized by an arching of the back and an immobile posture by the female in response to male mounting. We found consistent activation of hypothalamic neurons upon mating using c-Fos immunoreactivity as a marker (Supplementary Fig. 2) in

ovary intact, proestrous female mice. Specifically, ~30% of RP3V kisspeptin neurons displayed c-Fos immunoreactivity in intact females (unpaired $t$ test; $P = 0.022$; Fig. 2a and Supplementary Fig. 3) following this experimental paradigm. We next asked whether a single injection with Kp-10 is sufficient to stimulate lordosis behavior in female mice. We found that a sc Kp-10 injection in OVX+E females robustly stimulated lordosis behavior (paired $t$ test; $P < 0.001$; Fig. 2b). Likewise, an intracerebroventricular (icv) injection of Kp-10 (paired $t$ test; $P = 0.012$; Supplementary Fig. 4) also stimulated lordosis behavior in C57Bl6 (OVX+E+P) female mice (for details on hormone treatments, see Supplementary Table 1). Next, we analyzed lordosis behavior in $Kiss^{-/-}$ (OVX+E+P) females and observed strong deficits (Mann–Whitney $U$ test; $P = 0.029$; Fig. 2c), which could be reversed by a single sc injection of Kp-10 (paired $t$ test; $P \leq 0.01$; Fig. 2d). Consistent with this, acute ablation of ~70% of kisspeptin neurons by bilateral injection of an AAV encoding a Cre recombinase-dependent caspase 3 into the RP3V of adult female (OVX+E+P) KissIC mice also led to profound deficits in lordosis behavior (unpaired $t$ test; $P = 0.006$; Fig. 2e), which were reversible upon a single sc Kp-10 injection (paired $t$ test; $P = 0.03$; Fig. 2f).

To test whether an activation of RP3V kisspeptin neurons is sufficient to trigger lordosis behavior, we stereotaxically injected an AAV encoding a Cre-dependent channelrhodopsin (ChR2) bilaterally into the RP3V of female KissIC mice (Fig. 2g). Blue

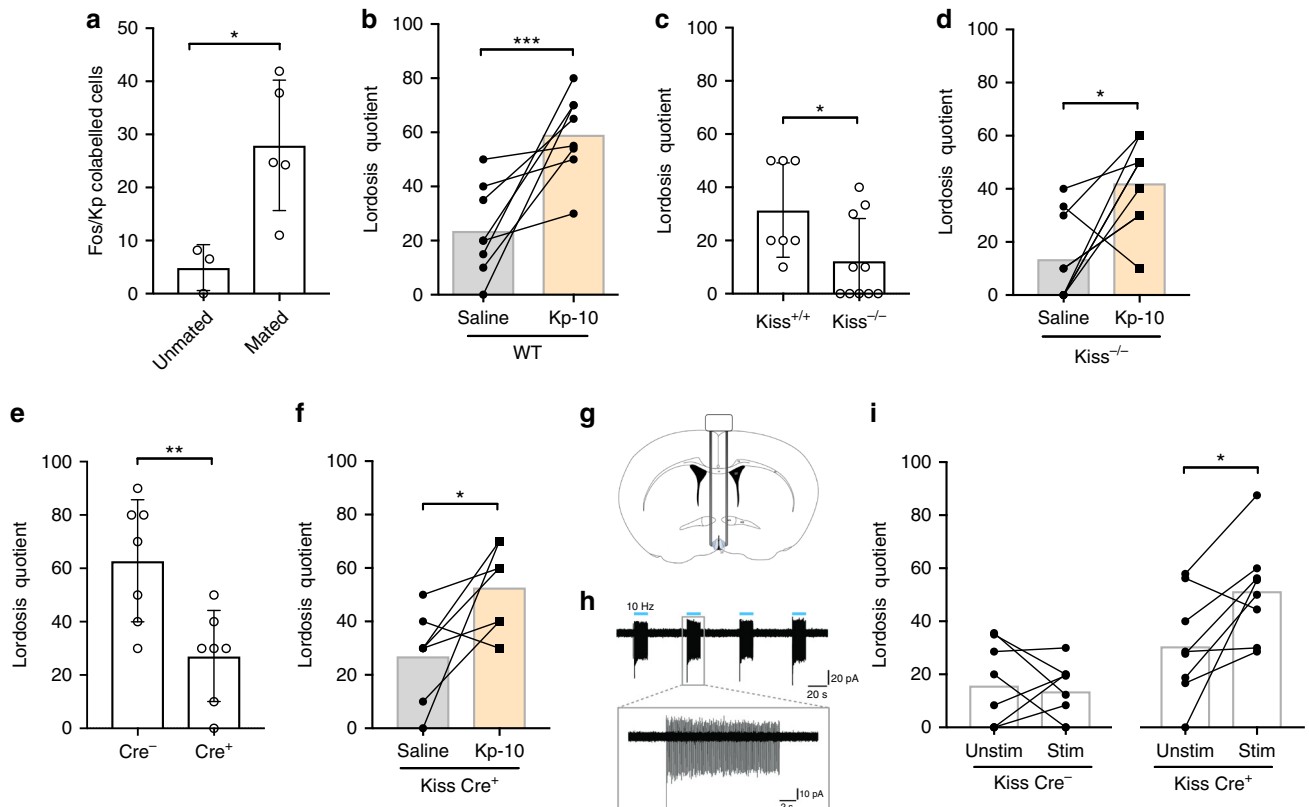

**Fig. 2** RP3V kisspeptin neurons are an important component of the neural network regulating lordosis behavior. **a** Mating specifically activated kisspeptin neurons in the RP3V of ovary intact female mice as determined by the percentage of Fos/Kp co-labeled cells. *$P \leq 0.05$; unpaired $t$ test; $n = 3/5$. **b** A peripheral injection of kisspeptin-10 (Kp-10) at a dose of 0.52 μg kg$^{-1}$ stimulated lordosis behavior. ***$P < 0.001$; paired $t$ test; $n = 8$ per group. **c** Lordosis behavior is attenuated in kisspeptin knockout (Kiss$^{-/-}$) mice. Please note that the background strain of Kiss$^{+/+}$ and Kiss$^{-/-}$ mice is 129SvJ, which showed overall lower levels of lordosis behavior compared to C57Bl/6J mice. *$P \leq 0.05$; Mann–Whitney $U$ test; $n = 7/10$. **d** A peripheral injection with Kp-10-induced lordosis behavior in Kiss$^{-/-}$ females. *$P \leq 0.05$; paired $t$ test; $n = 9$. **e–f** Stereotaxic injection with an AAV encoding a Cre-dependent caspase bilaterally into the RP3V decreased lordosis behavior, but was restored by a peripheral Kp-10 injection. **e****$P \leq 0.01$; unpaired $t$ test; *$P \leq 0.05$. **f** Paired $t$ test; $n = 7$ per group. **g** Anatomical drawing showing the position of the bilateral cannula holding optical fibers with 45° oriented mirrors tip into the RP3V. **h** Blue light photostimulation (10 Hz, 473 nm) elicited robust firing of kisspeptin neurons in KissIC mice brain slices, which were injected with an AAV encoding a Cre-dependent channelrhodopsin (AAV-ChR2) bilaterally into the RP3V. **i** Blue light photostimulation (Stim) increased the expression of lordosis behavior in KissIC mice, which were injected with AAV-ChR2 bilaterally into the RP3V. *$P \leq 0.05$, paired $t$ test; $n = 8$ per group (Kiss Cre$^-$ and Kiss Cre$^+$). Bars represent the mean ± SEM. For all experimental details, see Supplementary Table 1

light photostimulation (1−15 s at 10 Hz) elicited robust firing in virally transduced kisspeptin neurons with spike fidelity of 99% (Fig. 2h) in brain slice preparations. Photostimulation of RP3V kisspeptin neurons in vivo at 10 Hz for <15 s per male mount also was successful in enhancing lordosis behavior in Cre$^+$ (OVX+E) female mice (paired $t$ test; $P = 0.025$; Fig. 2i) without enhancing lordosis expression in Cre$^-$ (OVX+E) female mice injected with the AAV-ChR2 virus (paired $t$ test; $P = 0.72$; Fig. 2i). Taken together, our experiments demonstrate that RP3V kisspeptin neurons are an integral part of the neural network involved in both mate preference and lordosis behavior in female mice.

**Olfactory control of lordosis behavior.** Next, we determined the individual contribution of the vomeronasal pathway and the main olfactory system on RP3V kisspeptin neuron activation during sexual interaction with a male. VNO removal (Dunn's multiple comparison test; $P = 0.006$; Fig. 3a) but not the ablation of the MOE (Dunn's multiple comparison test following Kruskal–Wallis test; $P > 0.99$; Fig. 3a) disrupted lordosis behavior in C57Bl/6J (OVX+E+P) females (without affecting the male behavior toward the female; Dunn's multiple comparison test; VNOi/ZnSO4 vs. VNOi/Saline $P > 0.99$; VNOx/Saline vs. VNOi/Saline $P > 0.99$;

VNOx/ZnSO4 vs. VNOi/Saline $P > 0.99$; Supplementary Fig. 5) indicating that lordosis mainly depends on a functional vomeronasal pathway. While ablation of the VNO resulted in a dramatic attenuation in lordosis behavior, c-Fos expression in RP3V kisspeptin neurons remained significant (Dunn's multiple comparison test; $P = 0.01$ compared to controls). This might reflect activation through the MOE by specific volatile odors only secreted by the male when in direct contact with the female, since ablation of both the VNO and the MOE in mice completely eliminated RP3V kisspeptin neuron activation (Fig. 3b). Regardless, activation of RP3V kisspeptin neurons exclusively through the main olfactory system when in direct contact with the male seems to be insufficient to trigger lordosis. Taken together, our data reveal that RP3V kisspeptin neurons are an essential part of a motivational neural pathway that is triggered by male olfactory cues detected and processed predominantly through the vomeronasal pathway, ultimately leading to the female adapting a specific mating posture facilitating intromission.

**Mate preference, but not lordosis, is GnRH dependent.** RP3V kisspeptin neurons directly innervate GnRH neurons[24,25] and are implicated in generating the preovulatory LH surge[16,17,26–28].

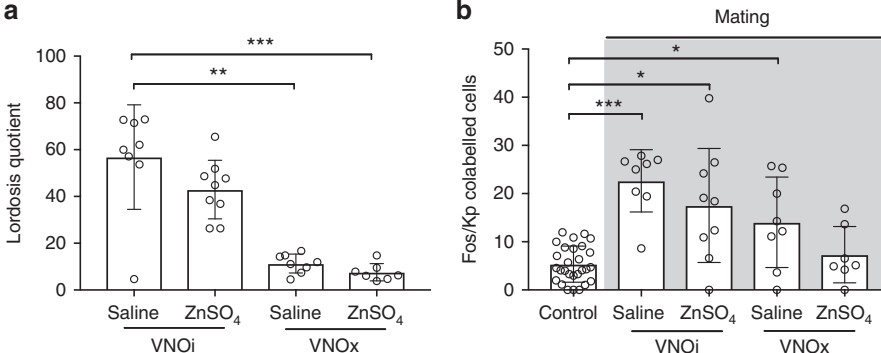

**Fig. 3** Lordosis behavior depends on the accessory olfactory system in female mice. **a** Lordosis behavior was strongly disrupted upon VNO removal, but only slightly after ablation of the MOE by infusion with a zinc sulfate solution (ZnSO₄). **b** Mating failed to activate RP3V kisspeptin neurons when both peripheral olfactory sensory input organs (VNOx/ZnSO₄) were ablated. Sham procedures were performed for each intervention as controls. No significant differences between control animals were found; therefore, all controls were combined into a single group. Bars represent the mean ± SEM. *$P \leq 0.05$; **$P \leq 0.01$; ***$P \leq 0.001$; Dunn's multiple comparison test; **a** $n = 8/9/8/7$; **b** $n = 8/9/8/7$. For all experimental details, see Supplementary Table 1

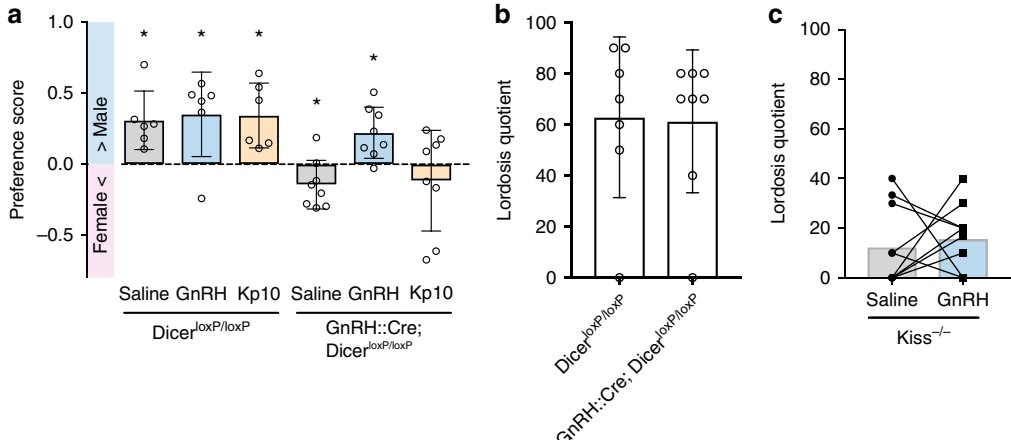

**Fig. 4** Mate preference, but not lordosis, behavior depends on GnRH signaling. **a** Genetic disruption of Dicer in GnRH neurons abolishing GnRH expression in the *GnRH::Cre; Dicer*$^{loxP/loxP}$ mouse model induced a female- instead of a male-directed preference, whereas control littermates showed a preference for the male (saline condition). A single peripheral GnRH injection at a dose of 0.025 mg kg$^{-1}$ induced a male-directed preference in *GnRH::Cre; Dicer*$^{loxP/loxP}$ female mice, whereas a peripheral injection with kisspeptin (Kp10) was not successful. *$P \leq 0.05$; one-sample *t* test (H0: mean equals 0); $n = 6$ (*Dicer*loxP/loxP) or 8 (*GnRH::Cre;Dicer*loxP/loxP). **b** Strikingly, such disruption of GnRH expression in *GnRH::Cre; Dicer*$^{loxP/loxP}$ mouse model did not affect lordosis behavior. Mann–Whitney *U* test; $P = 0.79$; $n = 7/8$. **c** A single injection with GnRH failed to stimulate lordosis behavior in *Kiss*$^{-/-}$ mice. Paired *t* test; $P = 0.65$; $n = 10$. Bars represent the mean ± SEM. For all experimental details, see Supplementary Table 1

Kisspeptin can activate GnRH neurons via its canonical receptor Kiss1R, which is expressed in ~95% of these cells[14,15,29]. To test whether RP3V kisspeptin neurons act on GnRH neurons to drive olfactory mate preference and lordosis behavior, we used *GnRH:: Cre; Dicer*$^{loxP/loxP}$ females, which are incapable of synthesizing and secreting GnRH in adulthood[30]. *GnRH::Cre; Dicer*$^{loxP/loxP}$ (OVX+E+P) females failed to show male-directed preferences and actually showed a preference for the female (one-sample *t* test (H0: mean equals 0); $P = 0.049$; Fig. 4a). A single sc injection of GnRH (0.025 mg kg$^{-1}$) restored this behavior in these females supplemented with only estradiol (OVX+E) (one-sample *t* test (H0: mean equals 0); $P = 0.01$; Fig. 4a), whereas an sc injection of Kp-10 failed to elicit a male-directed preference in female (OVX +E+P) *GnRH::Cre; Dicer*$^{loxP/loxP}$ mice (one-sample *t* test (H0: mean equals 0); $P = 0.38$; Fig. 4a). We found that lordosis behavior was not affected in (OVX+E+P) *GnRH::Cre; Dicer*$^{loxP/loxP}$ females (Mann–Whitney *U* test; $P = 0.79$; Fig. 4b). These results suggest that although GnRH neurons are required for mate preference, they might not be essential for the expression of lordosis behavior. Consistent with this, GnRH injection into

*Kiss*$^{-/-}$ females failed to stimulate lordosis behavior in (OVX+E) mice (paired *t* test; $P = 0.65$; Fig. 4c).

**Kisspeptin control of lordosis is mediated by nitric oxide**. To identify potential candidate neurons downstream of RP3V kisspeptin neurons other than GnRH neurons, we used ovary-intact female *KissIC/R26-BIZ* mice which express the transsynaptic tracer barley lectin (BL) exclusively in kisspeptin neurons[31]. Because BL is also expressed in ARC kisspeptin neurons in these animals, we injected a Cre-dependent mCherry adeno-associated virus bilaterally into the RP3V to delineate the projections from RP3V kisspeptin neurons. We observed a cluster of BL+ cells in the ventrolateral part of the ventromedial hypothalamus (VMHvl), a brain area previously implicated in reproductive behaviors[32]. Subsequent immunohistochemical analyses (Fig. 5a–g) showed that a major subpopulation of the BL+ neurons in the VMHvl express neuronal nitric oxide synthase (nNOS), previously implicated in reproductive behaviors[33,34]. Furthermore, our tract tracing suggests that RP3V kisspeptin

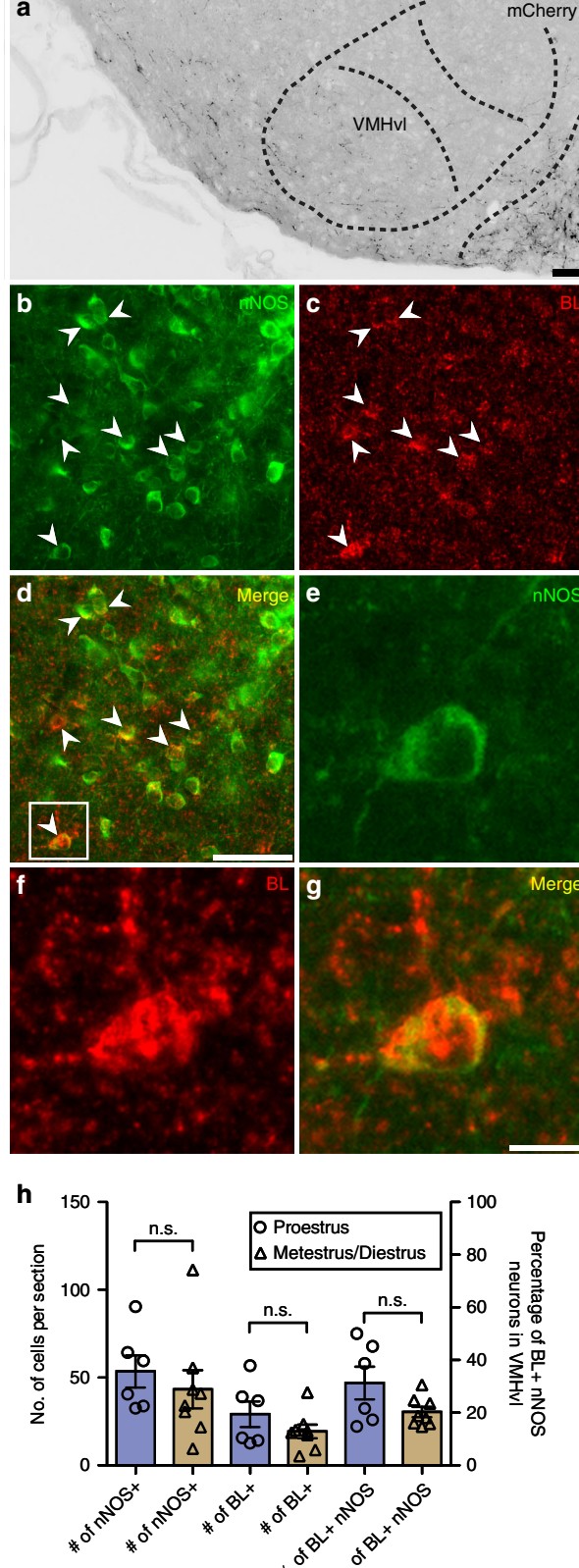

**Fig. 5** VMHvl nNOS neurons are connected to kisspeptin neurons. **a** mCherry-immunoreactive projections were detected in the VMHvl after injection of AAV-ChR2 into the RP3V of *KissIC/R26-BIZ* mice (scale bar = 50 μm); n = 3. **b–d** Transsynaptic tracing reveals that nNOS neurons in the VMHvl are (either directly or indirectly) connected to kisspeptin neurons (scale bar = 50 μm). BL+ nNOS neurons are indicated with arrows. **e–g** Zoomed-in image of insert shown in **d** (scale bar = 10 μm). **h** The number of neurons expressing nNOS in the VMHvl, the number of BL+ cells, and the overall percentage of BL+ nNOS neurons were not found to be significantly different (Bonferroni's multiple comparison test) between proestrus and metestrus/diestrus. n = 6 for proestrus, n = 8 for metestrus/diestrus. Bars represent the mean ± SEM. For all experimental details, see Supplementary Table 1

significant ($P > 0.05$, Bonferroni multiple comparison test; Fig. 5h). Taken together, these data indicate that nNOS neurons within the VMHvl are part of a neural pathway containing kisspeptin neurons.

To further dissect the functional role of NO signaling in this neural circuit important for sexual behavior, we next analyzed mate preference and lordosis behavior in mice deficient in nNOS. We found that nNOS knockout ($nNOS^{-/-}$) (OVX+E+P) females actually showed a small, albeit significant, preference for the female (one-sample *t* test (H0: mean equals 0); $P = 0.01$; Fig. 6a). However, a male-directed preference comparable to control littermates could be induced when injected sc with a cocktail of the nitric oxide donor SNAP and the guanylate cyclase agonist BAY 41-2272 (one-sample *t* test (H0: mean equals 0); $P = 0.03$; Fig. 6a). Accordingly, $nNOS^{-/-}$ (OVX+E+P) females also showed a strong decrease in lordosis behavior compared to control littermates (Tukey's multiple comparison test; $P = 0.002$; Fig. 6b), which was restored by an sc injection of the cocktail SNAP+BAY 41-2272 (Tukey's multiple comparison test; $P = 0.04$; Fig. 6b). By contrast, an sc injection of either KP-10 into female (OVX+E+P) mice or GnRH into (OVX+E) mice failed to stimulate lordosis behavior in $nNOS^{-/-}$ females (Tukey's multiple comparison test; respectively $P = 0.56$; $P = 0.19$; Fig. 6c). In a final experiment, $Kiss^{-/-}$ (OVX+E+P) females were injected sc with the cocktail SNAP+BAY 41-2272 and WT-like levels of lordosis behavior were observed (paired *t* test; $P = 0.02$; Fig. 6d).

Taken together, these data demonstrate that NO is a key neurotransmitter downstream of kisspeptin neurons mediating both mate preference and lordosis behavior. The nNOS neuron population in the VMHvl might be a potential important downstream relay of RP3V kisspeptin neurons in governing lordosis behavior but also mate preference consistent with previous studies[35].

## Discussion

The kisspeptin peptide is a well-established potent activator of the hypothalamic-pituitary-gonadal axis. The present study provides the first evidence that kisspeptin neurons in the RP3V are an essential part of a motivational neural circuit that is triggered by male olfactory cues detected and processed predominantly through the vomeronasal pathway, ultimately leading to the female adapting the lordosis posture facilitating intromission and consequently fertilization. Our data establish these cells as a central hub in the neural network governing the orchestration of sexual behavior in female mice.

Mate preferences are sexually differentiated: when in breeding condition, males and females seek out and mate with opposite sex conspecifics[20]. Many mammalian species use olfactory cues to identify potential mates[3]. It has therefore been postulated that sex differences in the circuits detecting and processing these olfactory

neurons project to the VMHvl and therefore, these VMHvl nNOS neurons may be downstream of RP3V kisspeptin neurons. We observed at proestrus that 31.27 ± 6.22% of nNOS neurons in the VMHvl were BL+, whereas at diestrus 20.25 ± 1.94% contained the tracer, however this was not found to be statistically

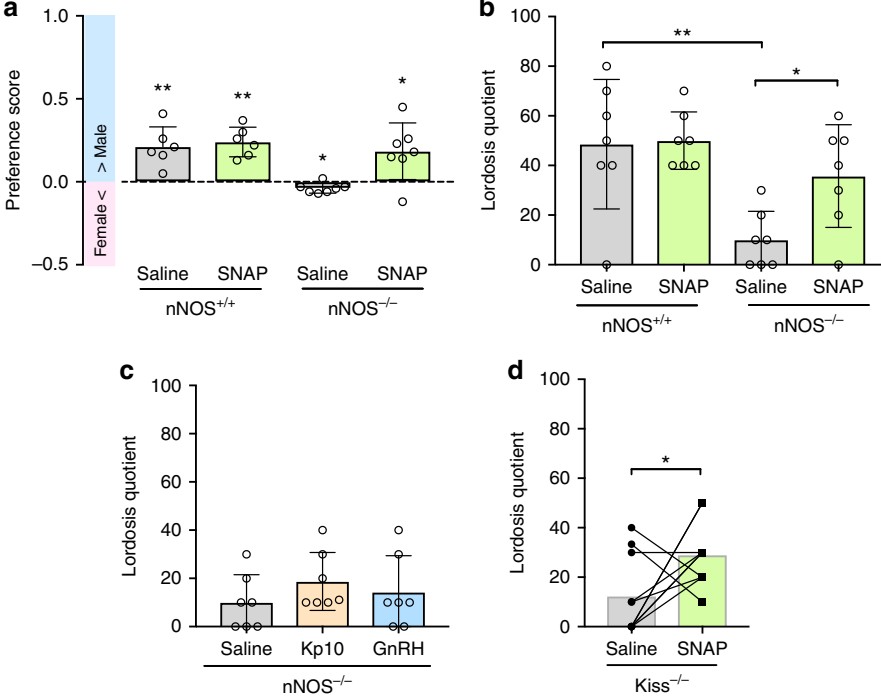

**Fig. 6** Mate preference and lordosis behavior depend on nitric oxide signaling. **a** nNOS knockout ($nNOS^{-/-}$) female mice do not show a male-directed preference, whereas female control littermates displayed a preference for the male. Strikingly, a single peripheral injection with SNAP+BAY 41-2272, NO donor and soluble guanylate cyclase agonist induced a significant preference for the male in $nNOS^{-/-}$ female mice. *$P \leq 0.05$; **$P \leq 0.01$; one-sample $t$ test (H0: mean equals 0); $n = 6$ ($nNOS^{+/+}$) and 7 ($nNOS^{-/-}$). **b** Lordosis behavior is disrupted in nNOS knockout ($nNOS^{-/-}$) mice, but restored by a peripheral injection of the NO donor SNAP (together with BAY 41-2272); *$P \leq 0.05$; **$P \leq 0.01$; Tukey's multiple comparison test; $n = 7$ per group. **c** By contrast, a peripheral injection of either kisspeptin or GnRH failed to restore lordosis behavior in $nNOS^{-/-}$ mice; ANOVA; $n = 7$ per group. **d** A peripheral injection with SNAP+BAY 41-2272-induced lordosis behavior in Kiss$^{-/-}$ mice; *$P \leq 0.05$; paired $t$ test; $n = 10$ per group. nNOS the neuronal form of nitric oxide synthase, Kp-10 kisspeptin, SNAP S-nitroso-N-acetylpenicillamine. Bars represent the mean ± SEM. For all experimental details, see Supplementary Table 1

cues might provide an entry point into understanding the neural basis of mate preferences. Indeed, numerous studies (reviewed in ref. [20]) using c-Fos as neuronal marker of activation have reported important sex differences along both the main and accessory olfactory pathways when animals are exposed to olfactory cues. However, how these olfactory stimuli are ultimately integrated in the neuroendocrine brain to trigger the appropriate behavioral and hormonal responses has not yet been elucidated.

Intriguingly, GnRH neurons receive direct inputs from both the main and accessory olfactory pathways, but none of these inputs were found to differ between the two sexes[9]. This suggests that additional neuronal populations must be involved in transferring chemosensory information to GnRH neurons in female mice. Our previous observations[19] of a sex-specific activation of the RP3V kisspeptin neuronal population by opposite sex odor cues suggest that this particular subset of neurons might present the missing link in how olfactory cues can elicit sexually differentiated behavioral and neuroendocrine responses. The present experiments now confirm that kisspeptin neurons in the RP3V indeed present a central hub in transferring olfactory information perceived through the vomeronasal pathway to the reproductive center of the neuroendocrine brain leading to sex-specific mate preferences and mating behaviors.

Kisspeptin acts through its canonical receptor Kiss1R expressed by ~95% of GnRH neurons to control GnRH neuronal activity and secretion. Previous studies have provided evidence for a direct role of GnRH in sexual behavior. Systemic or brain injections with GnRH can stimulate sexual behavior in both male and female rodents[36–39], suggesting that GnRH neurons are part of the neural circuits regulating these behaviors. Consistent with

this, it was shown that GnRH neurons have synaptic connections with subsets of neurons in several brain areas previously implicated in sexual behavior using a mouse model in which the transneuronal tracer barley lectin was specifically expressed in GnRH neurons[9]. These include the medial amygdala and the posteromedial part of the bed nucleus of the stria terminalis, both known to relay pheromone signals, as well as the medial preoptic area, the ventrolateral part of the ventromedial hypothalamus (VMHvl) and the ventral part of the premammilary nucleus (PMV). Using a viral-mediated tract tracing approach, it was demonstrated that kisspeptin neurons in the RP3V directly contact GnRH neuron cell bodies and proximal dendrites in the rostral preoptic area[24]. It thus seemed reasonable to assume that kisspeptin's effects on sexual behavior are mediated through Kiss1R on GnRH neurons. Intriguingly, $GnRH::Cre; Dicer^{loxP/loxP}$ females showed normal levels of lordosis behavior upon priming with estradiol and progesterone. These data are consistent with previous studies in mice harboring a deletion in the GnRH gene (and therefore fail to synthesize GnRH), which also appear (after OVX and priming with estradiol and progesterone) to have no deficits in sexual behavior[40].

RP3V kisspeptin neurons, in contrast, seem to act through GnRH signaling in modulating mate preferences, since mate preferences were clearly disrupted in $GnRH::Cre; Dicer^{loxP/loxP}$ females. Furthermore, a peripheral injection with GnRH, but not with Kp-10, restored a male-directed preference in $GnRH::Cre; Dicer^{loxP/loxP}$ females, suggesting that GnRH neurons must be downstream of kisspeptin neurons in this neural circuit. We found that lordosis behavior was normal in $GnRH::Cre; Dicer^{loxP/loxP}$ females. This result is in line with a previous study[41] showing normal lordosis behavior in female Kiss1R knockout mice.

Several studies[42,43] have demonstrated that kisspeptin can activate neuropeptide FF receptors (NPFFR1 and NPFFR2). Herbison and colleagues[44] have demonstrated that kisspeptin can modulate arcuate neuron excitability at least partially via NPFF receptors independently of Kiss1R. As NPFFR2 and (to a lesser extent) NPFFR1 have been detected in the VMH of mice[45], it is tempting to speculate that these receptors may play a key role in the facilitation of lordosis behavior. Future studies using NPFF receptor mutant mice may provide new evidence of the involvement of these receptors in the kisspeptidergic modulation of reproductive behavior.

Taken together, our genetic analyses challenge the idea that GnRH plays a prominent role in lordosis behavior under normal physiological conditions. There are some caveats that we must consider however regarding the *GnRH::Cre; Dicer^loxP/loxP* mouse model used in the present studies. Although there is no longer any GnRH synthesis and secretion in adult *GnRH::Cre; Dicer^loxP/loxP* mice[30], these neurons do not die and remain electrically active and could thus presumably still respond to kisspeptin through Kiss1R. Nevertheless, the present results suggest that GnRH neurons rather play a facilitatory than a critical role in lordosis behavior. Finally, it is quite likely that specialized subpopulations of kisspeptin neurons exist within the RP3V since transynaptic tracing has indicated that the majority of kisspeptin neurons within the RP3V does not communicate with GnRH neurons[46]. This suggests that RP3V kisspeptin neurons may have other functions in the adult female mouse brain that do not directly involve GnRH neurons and this could well be mediating olfactory-driven sexual behavior.

We detected BL+ cells in the VMHvl, a brain area necessary for the expression of lordosis behavior in *KissIC/R26-BlZ* mice, in which BL is specifically expressed in kisspeptin neurons[31]. Furthermore, we identified a major subpopulation of these neurons as nNOS+. While no significant difference was found between the percentage of BL+ nNOS neurons at proestrus compared to metestrus/diestrus, we noticed that the BL+/nNOS+ cluster in the VMHvl was more clearly detected when females were in proestrus. Previous studies have indicated that WGA (which has ~98% sequence homology with BL) is activity dependent[47], however the half-life of BL within neurons remains unknown. Therefore, it is unclear whether this genetic model provides sufficient temporal resolution to accurately detect changes in synaptic activity across the limited time frame of the estrous cycle.

The trend toward estrous cycle-dependent changes in BL/nNOS immunoreactivity could, however, be potentially interesting in light of the recent study by Dey and colleagues[48] showing that female mouse vomeronasal sensory neurons (VSN) were temporarily silenced to a subset of male pheromones during diestrus, but were fully responsive to the same odors when in estrus. It was proposed that the silencing of these VSN occurred under the influence of progesterone, which temporarily rises during diestrus. In this respect, it is interesting to note that RP3V kisspeptin neurons also express progestin receptors and thus are sensitive to the actions of progesterone. Furthermore, progestin receptors are also present in the lateral part of the VMH and its expression is highly dependent on estrogens[49–52]. Finally, a recent study[53] showed that close to 100% of neurons expressing estradiol receptors in the VMHvl are actually nNOS-expressing neurons. Taken together, these findings reveal that kisspeptin neurons communicate with nNOS neurons in the VMHvl, a central hub driving female sexual behavior. Further specific ablation or stimulation of the VMHvl nNOS population will be necessary to decipher its precise role in female reproductive behaviors.

Several studies have implicated the neurotransmitter NO in sexual behavior[33,34]. The present studies show that nNOS neurons are most likely downstream of kisspeptin and GnRH

neurons since neither GnRH nor Kp-10 was able to induce lordosis behavior in *nNOS^−/−* females. Possibly the nNOS neuron population in the VMHvl plays an important role because of its synaptic connection with kisspeptin neurons. However, it cannot be ruled out that other nNOS neuron populations are also involved, for instance, in the rostral preoptic area, where nNOS neurons are part of a neural circuitry mediating ovarian cyclicity and ovulation[54], and in the paraventricular nucleus (PVN), where nNOS co-localizes with oxytocin. The latter has also been implicated in sexual behavior in female rodents: mating activates PVN oxytocin neurons and increases local oxytocin concentrations, in particular following a pace mating paradigm in which the female controls the mating, thereby linking oxytocin in particular with sexual motivation[55,56]. Therefore, further experiments are needed to dissect the precise role of the different nNOS populations in the brain in controlling reproductive behaviors.

Taken together, our data strongly suggest that RP3V kisspeptin neurons mediate sexual behavior and that hypothalamic NO signaling could be an essential mechanism downstream of kisspeptin neurons in governing sexual behavior.

The discovery that a neuropeptide that governs ovulation is also able to elicit sexual behavior opens new avenues of research for the treatment of low sexual desire in women, including hypoactive sexual desire disorder (HSDD). HSDD represents the most severe form of low sexual desire. Although the causes of low sexual desire and HSDD are multi-factorial, there is convincing evidence that sex hormones play an important role since the prevalence of HSDD increases after natural or surgically induced menopause[57]. At present, there is no good treatment available for low sex drive including HSDD in women. Testosterone has been used in postmenopausal women but also has strong adverse effects leading to virilization. For premenopausal women, there is currently no treatment available. A recent study from Comninos et al.[58] used functional magnetic resonance imaging to show that kisspeptin enhances limbic brain activity specifically in response to sexual and couple-bonding visual stimuli in men. Taken together, therapeutics targeting kisspeptin signaling in the brain may hold the potential to provide novel treatment options for low sexual desire.

## Methods

**Mouse models.** Kisspeptin knockout (*Kiss^−/−*)[22], Kisspeptin-IRES-Cre (*KissIC*)[26], *GnRH::Cre*[12]; *Dicer^loxP/loxP*[59], *R26-BIZ*[31], and nNOS knockout (*nNOS^−/−*)[54,60] mouse strains have been previously described and validated, and are on different genetic backgrounds (see Supplementary Table 1). All experiments were performed on adult (>8 weeks of age) female mice unless otherwise stated.

Animal care and experimental procedures were performed in accordance with the guidelines established by the institutional animal care and use committee of the Royal Netherlands Academy of Arts and Science and by the National Institutes of Health "Guide for the Care and Use of Research Animals, Eight Edition," and were approved by the Ethical Committee for Animal Use of the Universities of Liege (Belgium), Saarland (Germany) and of Otago (New Zealand). Female mice were placed into individual cages under a reversed light/dark cycle (12 h:12 h light/dark; 21.00 h lights on and 9.00 lights off) with food and water ad libitum.

**Ovariectomy and hormone supplementation.** Unless otherwise stated, females were ovariectomized in adulthood (>8 weeks of age) under general anesthesia after either subcutaneous (sc) injections of ketamine (80 mg kg^−1 per mouse) and medetomidine (Domitor, Pfizer, 1 mg kg^−1 per mouse) or under 5% isoflurane, in order to control for endogenous hormone concentrations and to prevent pregnancies upon repeated testing. At the same time, all females received a 5-mm-long silastic capsule (inner diameter: 1.57 mm; outer diameter: 2.41 mm) containing crystalline 17β-estradiol (diluted 1:1 with cholesterol) subcutaneously in the neck. The dose of E₂ (E8875, Sigma) was based on a previous study[61] showing that this treatment leads to estradiol levels similar to mice in estrus. At the end of surgery, females under ketamine/medetomidine anesthesia received an sc injection of atipamezole (Antisedan, Pfizer, 4 mg kg^−1 per mouse) to antagonize medetomidine-induced effects and accelerate recovery. In order to induce sexual receptivity at the day of testing, all females received a subcutaneous injection with progesterone (500 μg, P0130, Sigma) 3 h before the onset of the behavioral test, unless stated otherwise (for overview of all the different hormone treatments, see Supplementary Table 1).

**Viruses and stereotaxic injections for behavioral testing**. The AAV5-flex-taCasp3-TEVp virus (abbreviated to "AAV-Casp3" (Vector Core, University of North Carolina)) was stereotaxically injected bilateral into the RP3V in *KissIC* mice (total: $Cre^-$: $n = 14$; $Cre^+$: $n = 14$). AAV-Casp3 uses the T2A peptide encoding sequence to ensure bicistronic expression of pro-taCasp3 and TEVp after Cre-mediated recombination. taCasp3 triggers cell autonomous apoptosis, thereby minimizing toxicity to adjacent $Cre^-$ cells[23].

AAV5-EF1a-DIO-hChR2(H134R)-mCherry-WPRE-pA virus (abbreviated to "AAV-ChR2" (Vector Core, University of North Carolina)) was stereotaxically (see below) injected bilaterally into the RP3V in *KissIC* mice ($Cre^-$: $n = 8$; $Cre^+$: $n = 8$) to selectively express the light-activated cation channel channelrhodopsin-2 and mCherry[62] in kisspeptin neurons.

Mice were placed in a motorized stereotaxic frame (Neurostar, Germany) under 5% isoflurane anesthesia. The skull was exposed by a midline scalp incision, and the stereotaxic frame was aligned at Bregma using visual landmarks. After alignment of the head of the mice, a drill was placed above the skull at coordinates (according to the Paxinos Brain Atlas[63]) corresponding to the rostral periventricular area of the third ventricle (RP3V; rostrocaudal, 0.2 mm; mediolateral, $\pm 0.1$ mm) and a hole drilled through the skull bone to expose the brain. A 33-gauge steel needle loaded with virus (AAV-Casp3 or AAV-ChR2) was slowly inserted through the hole until it penetrated to a depth of 5.8 mm. Virus (1 µl per brain site injected) was delivered at 100 nl min$^{-1}$ through a Hamilton syringe using a syringe pump (Harvard Apparatus). The needle was left in place for an additional 10 min to allow diffusion of the virus before being slowly removed. Following AAV-Casp3 injection, the hole was filled with dental cement and the skin was sutured. Following AAV-ChR2 injection, a bilateral cannula (200 µm core diameter; Doric Lenses) holding optical fibers with 45°-oriented mirror tips was inserted into the RP3V at a distance of 0.25 mm (mediolateral) from the center of injection and further fixed to the skull with dental cement. Mice were allowed to recover on a heating pad and returned to their home cage after waking up. All mice received an sc injection with Caprofen (5 mg kg$^{-1}$) for post-operative analgesia.

**Viruses and stereotaxic injections for electrophysiology**. Adult female (>2 months old) heterozygous *KissIC* mice were group-housed under conditions of controlled temperature ($22 \pm 2$ °C) and lighting (12-h light, 12-h dark cycles) with ad libitum access to food and water. Mice were anesthetized, placed in a stereotaxic apparatus, and given simultaneous bilateral 0.5 µl injections of AAV9-EF1-DIO-hChR2-(H134R)-mCherry-WPRE-hGH ($2.2 \times 10^{13}$ GC ml$^{-1}$; Penn Vector Core) into the RP3V (coordinates according to the Paxinos Brain Atlas[63]), 0.2 mm anterior to Bregma and 5.8 mm in depth) at a rate of 100 nl min$^{-1}$. The syringes were left in situ for 3 min before and 10 min after the injections. Following a recovery period, mice were bilaterally ovariectomized under anesthesia and, after >2 weeks, received subcutaneous silastic implants containing 17-β-estradiol (1 µg per 20 g body weight) according to Bronson[64]. Implants were made of 17-β-estradiol dissolved in ethanol and mixed with medical grade adhesive (0.1 mg ml$^{-1}$ adhesive), which is then injected into 1 mm internal diameter silastic tubing. Six days later, mice received a subcutaneous injection of estradiol benzoate (1 µg per 20 g body weight) in the morning and were used for electrophysiology the following day

**Cannula implantation for ICV kisspeptin administration**. Mice were placed in a motorized stereotaxic frame (Neurostar, Germany) under 5% isoflurane anesthesia. The skull was exposed by a midline scalp incision, and the stereotaxic frame was aligned at Bregma using visual landmarks. After alignment of the head of the mice, a drill was placed above the skull at coordinates corresponding to the lateral ventricle (lateral+1, anterior–posterior: −0.34; dorsoventral: −2.5) and a hole drilled through the skull bone to expose the brain. Then, a 26-gauge cannula cut at 2 mm from pedestal was implanted and fixed to the skull with dental cement. A dummy was inserted to close the cannula until the behavioral experiment. Mice were allowed to recover on a heating pad and returned to their home cage after waking up. All mice received an sc injection with Temsegic (0.05 mg kg$^{-1}$) for post-operative analgesia.

**Removal of the VNO**. One week after ovariectomy, subjects underwent either bilateral removal of the VNO or sham surgery (VNOx or VNOi groups)[65]. Briefly, animals were placed on their back and the lower jaw was gently opened after general anesthesia. A midline incision was made in the soft palate extending rostrally from behind the first palatal ridge to the incisors, and the underlying bone was exposed by blunt dissection. In VNOi animals, the incision was then closed with reabsorbable sutures. For VNOx animals, the rostral end of the VNO was exposed by drilling, the caudal end of the vomer bone was cut, and the VNO was removed bilaterally with a gentle twisting motion. Bleeding was controlled using a blunted 18-gauge needle attached to a vacuum. Animals were carefully monitored after surgery for bleeding and or breathing difficulties.

**Ablation of the MOE**. Two weeks after the removal of the VNO (VNOx females) or sham surgery (VNOi females), mice received an intranasal application of 10% ZnSO$_4$ to lesion the main olfactory epithelium (MOEx) or saline solution (MOEi) under general anesthesia[66].

**Kisspeptin treatment**. Kisspeptin-10 (Kp-10) was synthesized in Strasbourg, France (Sequence: Tyr-Asn-Trp-Asn-Ser-Phe-Gly-Leu-Arg-Tyr-NH2, NeoMPS; weight = 25.7 mg). To determine whether kisspeptin-stimulated female sexual behavior in wild-type *C57BL/6J* mice, females received an sc injection of Kp-10 at the dose of 0.52 µg kg$^{-1}$ (injection volume 100 µl) 2 h before the lordosis test. Each animal was used as its own control, i.e., the animal was injected 1 day with Kp-10 and the other day with saline. Females were not injected with progesterone in this particular experiment. Injections were separated by at least 3 days.

When injected intracerebroventricularly, females were injected with 10.4 ng kg$^{-1}$ of Kp-10 (injection volume 2 µl), 1 h before the lordosis test through a cannula inserted into the lateral ventricle.

**GnRH treatment**. Two hours before mate preference test, female mice received a single sc injection of GnRH (0.025 mg kg$^{-1}$, Polypeptide Laboratories France SAS, SC087).

**SNAP treatment**. S-nitroso-N-acetyl-DL-Penicillamine (SNAP) (N398-Sigma) is a NO donor. In order to ascertain the most efficient activity of SNAP (8 mg kg$^{-1}$) was combined with the guanylate cyclase agonist BAY-41-2272 (10 mg kg$^{-1}$; B8810 —Sigma) 1 h before injection. One hour before behavioral tests, female mice received a single subcutaneous injection (100 µl) of the cocktail SNAP+BAY 41-2272.

**Brain slice preparation for electrophysiology**. Mice were killed by cervical dislocation, decapitated and brains quickly removed. Coronal brain slices (200–250 µm) containing the rostral periventricular area of the third ventricle (RP3V) were cut with a vibratome (VT1000S; Leica) in an ice-cold solution containing (in mM): NaCl 87, KCl 2.5, NaHCO$_3$ 25, NaH$_2$PO$_4$ 1.25, CaCl$_2$ 0.5, MgCl$_2$ 6, glucose 25, and sucrose 75. Slices were then incubated at 30 °C for at least 1 h in artificial cerebrospinal fluid (aCSF; in mM): NaCl 120, KCl 3, NaHCO$_3$ 26, NaH$_2$PO$_4$ 1, CaCl$_2$ 2.5, MgCl$_2$ 1.2, and glucose 10. All solutions were equilibrated with 95%O$_2$/5%CO$_2$.

**Cell-attached recordings and light stimulation**. Slices were placed under an upright microscope fitted for epifluorescence (Olympus, Tokyo, Japan) and constantly perfused (1.5 ml min$^{-1}$) with warm (~30 °C) aCSF. mCherry-expressing RP3V neurons were first visualized by brief fluorescence illumination and subsequently approached using infrared differential interference contrast optics. Action potential firing was recorded in voltage clamp mode in the cell-attached loose patch configuration. Recording electrodes (3–5 MΩ) pulled from borosilicate capillaries (Warner Instruments, Hamden, CT) with a horizontal puller (Sutter Instruments, Navato, CA) were filled with aCSF including 10 mM HEPES. Low resistance seals (10–30 MΩ) were achieved by applying either no suction or the lowest amount of suction required to detect spikes. For ChR2 activation, blue light was delivered to the slice through a ×40 immersion objective (0.8 NA, Olympus) via a 470 nm light-emitting diode (LED, CoolLED) connected to the vertical illumination port of the microscope. Stimulation consisted of 1–15 s trains of blue light pulses (2 ms duration; ~0.25 mW) delivered at 10 Hz, repeated ten times every 60 s in each cell. Electrophysiological signals were recorded using a Multiclamp 700B amplifier (Molecular Devices, Sunnyvale, CA) connected to a Digidata 1440A digitizer (Molecular Devices). Signals were low-pass filtered at 3 kHz before being digitized at a rate of 10 kHz and stored on a personal computer. Signal acquisition and analysis was carried out with pClamp 10 (Molecular Devices). Spikes were detected using the threshold crossing method. In each cell, spike fidelity was calculated by dividing the number of light-evoked spikes by the number of blue light stimuli and expressed as a percentage.

**Behavioral tests**. All experimental females were brought into behavioral estrus by ovariectomy (OVX) in adulthood and combined treatment with estradiol (E) through a silastic capsule and an acute injection (3 h before testing) with progesterone (P), unless stated otherwise (for all details on hormone treatments, see Supplementary Table 1). Females were always tested during the dark phase of the light/dark cycle. Finally, levels of female sexual behavior displayed by the control (wild type) females vary as function of the background strain with *129SvJ* females ($Kiss^{+/+}$) showing relatively low levels compared to *C57BL6/j* females.

**Assessment of the MOE lesion**. Anosmia was assessed by submitting females to the hidden cookie test[66]. Briefly, female mice were food-deprived overnight. A small piece of a chocolate chip cookie was buried (~1 cm deep) at a random location in a clean Plexiglas aquarium (35 cm long × 25 cm high × 19 cm wide) containing fresh sawdust. The time it took each mouse to find the cookie was recorded. The test lasted until the mouse had located the cookie or 10 min if the cookie was not found. All mice treated with ZnSO$_4$ failed to find the hidden cookie and were thus considered to be anosmic.

**Exposure to odors in bedding**. Four groups of gonadally intact males ($n = 5$ each) were placed in clean cages containing fresh sawdust. Bedding was collected 12 h later and directly used as olfactory stimulus for the experimental females. Thirty-six hours before bedding exposure, all experimental females (singly housed) were

placed on clean sawdust in two separate housing units to separate females, which were going to be exposed to male bedding or to clean bedding as control (and thus to prevent the controls from being exposed to male odors). On the day of testing, females were injected with P (500 µg) to induce behavioral estrus. This hormonal treatment made the experimental females behaviorally receptive at the time of odor exposure. Three hours after P injection, 15 g of fresh male-soiled or clean bedding was placed into the subject's own cage. Ninety minutes after bedding exposure, females were perfused with paraformaldehyde and brains were collected.

**Mate preference tests**. To assess mate preferences shown in response to auditory and olfactory stimuli, we used a box (60 cm long × 30 cm high × 13 cm wide) that was divided into three compartments using perforated opaque partitions. The partitions contained perforated holes at a height of 8 cm to facilitate the diffusion of odors from the two-side compartments to the middle compartment. Tests were performed during the dark phase of the light cycle (5 h after lights out). Animals were habituated to the three compartment box only once on the day before the behavioral experiments by placing them in the middle compartment for 10 min (with no stimulus animals placed in the two-side compartments). On the day of testing, an intact male stimulus and an estrous female stimulus were placed in the lateral compartments with their own bedding to make the stimuli as odorous as possible. Three hours after receiving a progesterone injection (500 µg), the female subject was introduced into the middle compartment, and was observed for 10 min. The time the subject spent poking her nose through the holes of the partition or actively sniffing the bottom of the partition in front of the female vs. male stimulus animal was recorded. A preference score was calculated by dividing the time spent investigating the male compartment minus the time spent investigating the female compartment by the total time spent investigating both compartments. A positive value of the preference score indicates a mate preference directed toward the stimulus male, whereas a negative value indicates a mate preference directed toward the stimulus female (for details, see ref. [67]).

**Lordosis tests without photostimulation**. Females were subjected to weekly lordosis tests in a Plexiglas aquarium (37 cm long × 17 cm high × 21 cm wide). A sexually experienced male was placed alone in the aquarium and allowed to adapt for 15 min. Subsequently, 3 h after receiving a subcutaneous progesterone injection (500 µg, P0130, Sigma) to induce behavioral estrus, the lordosis responses of the female to the mounts of the stimulus male were recorded. The test lasted until the female received 10 mounts or 10 min had elapsed. For the first experiment (mating-induced Fos activation), ovary intact females were paired with males during 30 min. A lordosis quotient (LQ) was calculated by dividing the number of lordosis responses displayed by the female subjects by the number of mounts received (x100). Before each experimental condition (drug injection, cell ablation, and optogenetic stimulation), all females were subjected to at least three lordosis tests (with progesterone) in order to acquire sufficient sexual experience and thus a significant LQ. Tests were performed during the dark phase of the light cycle (5 h after lights out; for details, see ref. [67]). For all details on the different hormone treatments (estradiol vs. estradiol+progesterone), see Supplementary Table 1.

**Lordosis tests with photostimulation**. Prior to the lordosis test, the cannula was connected to an optical fiber (Doric Lenses), which in turn was connected to a blue laser (wavelength = 473 nm) via an optical rotatory join allowing free movements of the animal. The optic fiber was flexible and long enough to allow the female to freely behave and interact with the male. After three pretests, *KissIC* females (*Cre⁻* and *Cre⁺*) were then divided in two groups. On test 4, half of the females received optogenetic stimulation (stimulated), whereas the other half did not (unstimulated). On test 5 (conducted 1 week later), unstimulated females received an optogenetic stimulation while previously stimulated females did not, thus each female acted as her own control. Blue light was delivered through the optic cable at 10 Hz as soon as the male approached the female (sniffing and showing mount attempt). The duration of the stimulation varied as a function of the male, i.e., the time it took him to mount the female, however this was never longer than 15 s. Tests were performed over 10 min and the number of mounts was recorded as well as the number of female lordosis responses. Importantly, in order to observe possible stimulatory effects of photostimulation on lordosis behavior, females were not injected with progesterone before the lordosis test, and were thus only on estradiol treatment (by silastic capsule, previously described).

**Transcardial perfusion and OCT embedding for histology**. Female mice were anesthetized and perfused transcardially with saline followed immediately by 4% ice-cold paraformaldehyde. Brains were removed and postfixed in 4% paraformaldehyde for 2 h. Brains were then cryoprotected in 30% sucrose[68] in PBS and when sunken, were embedded in Optimal Cutting Temperature compound (OCT, Tissue-Tek). A glass box was placed in a slurry of ethanol and dry ice. The glass box was then partially filled with isopentane. The tissue was placed in a plastic cuvette filled with OCT, placed into the isopentane bath and rapidly frozen. Brains were subsequently stored at −80 °C prior to sectioning.

**Histological assessment of VNO removal**. As previously described[65], snouts were removed immediately after perfusion, cleared of all soft tissue, and soaked for

30 min in rapid decalcifier (Apex Engineering Products). Decalcified snouts were then soaked overnight in 30% sucrose[68] at which time a 1:1 mixture of 30% sucrose and OCT was suctioned into the nasal passages. Snouts were then incubated for 4 h in the 1:1 solution and were finally frozen in OCT and stored at −80 °C. Snouts were sectioned at 10 µm thickness on a cryostat. One section every 150 µm was transferred directly onto Superfrost Plus glass slides and dried overnight. Sections were rinsed and stained with hemotoxylin and eosin to assess the presence of blood clots in the nasal sinuses and to determine whether the VNO was completely removed. A total of 32 mice were used and upon examination, 8 were excluded because the VNO was not completely removed. No blood clots were detected in any of the animals.

**Immunohistochemical detection of c-Fos or kisspeptin**. Brain sections (30 µm thick) were cut on a Leica CM3050S cryostat. Forebrains were cut coronally from the rostral telencephalon to the posterior hypothalamus. Sections were saved in four different series, placed in antifreeze solution, and stored at −20 °C.

Immunostaining was carried out on free-floating sections. All incubations were carried out at room temperature, and all washes of brain tissue sections were performed using Tris-buffered saline (TBS 0.05 M) or Tris-buffered saline containing 0.1% Triton X-100 (TBST). Briefly, sections were rinsed and endogenous peroxidase activity was quenched by incubating the sections for 30 min with 0.3% hydrogen peroxide. Non-specific-binding sites were then blocked by incubating sections for 30 min with 5% normal goat serum (NGS) (Dako Cytomation, Denmark). Sections were then incubated either with a rabbit polyclonal anti-kisspeptin (1/5000 in TBST-NGS 5%; anti-kisspeptin-10, AB9754, Chemicon, Millipore) raised against the decapeptide kisspeptin-10 (derived from the *Kiss-1* gene product) for 48 h at 4 °C or with a rabbit polyclonal anti-c-Fos antibody (1/2000 in TBST-NGS 5%; c-Fos (4): sc-52R, Santa Cruz Inc.) raised against the N terminus of c-Fos of human origin. Sections were then incubated for 1 h in avidin-biotin complex (1/800, ABC, Vector Laboratory, Burlingam, CA) and then reacted for 5 min with 3,3′-diaminobenzidine tetrahydrochloride (DAB Kit, Vector Laboratory). Sections were then washed, dried overnight, left in xylene (Sigma) for 15 min, and coverslipped using Eukit (Fluka, Steinheim, Germany).

**Immunohistochemical detection of c-Fos and kisspeptin**. To determine the distribution of c-Fos and kisspeptin double-labeled neurons, ovary intact females in proestrus (activation of kisspeptin cells following mating) or ovariectomized females (VNOx/MOEx experiment) were perfused with 4% paraformaldehyde in 0.1 M PBS 90 min after the introduction of the male to the female (onset of behavioral testing) or 90 min after being placed in the empty testing arena for the unmated controls. Regarding male bedding exposure (VNOx/MOEx), females were perfused 90 min after the onset of odor exposure (male or clean bedding). For the dual immunohistochemistry, sections were first washed in 0.1 M PBS pH 7.4 (PBS), peroxidase activity was blocked in PBS solution with 0.3% $H_2O_2$, and then permeabilized in PBS-0.1% Triton X-100 (PBST) and saturated in 5% NGS in PBST. Immediately after this step, sections were incubated in diluted anti-c-Fos antibody overnight. On the following day, sections were washed in PBST and incubated in a goat anti-rabbit biotinylated secondary antibody (Dako, Prod. Ref. B0432, 0.75 µg ml⁻¹ PBST). Sections were then washed in PBST and incubated in the Vectastain Elite ABC Kit (Vector, Prod. Ref. PK6100). After development with the DAB Substrate Kit (Vector, SK-4100), in a black precipitate (3,3′-diaminobenzidine (DAB) plus Ni²⁺), sections were washed thoroughly in PBS, and residual peroxidase activity blocked in PBS solution with 0.3% $H_2O_2$. Sections were then permeabilized and blocked in 5% NGS-PBST and incubated in anti-kisspeptin-10 antibody for 72 h. Similar secondary antibody and ABC incubation steps were then performed. The developing reaction used in this step was a DAB brown precipitate, using the same kit. Following this, sections were mounted in Eukitt after being air-dried.

**Immunohistochemical detection of barley lectin and nNOS**. In order to identify cells that are synaptically connected to kisspeptin neurons, *KissIC/R26-BIZ* mice were transcardially perfused with 4% paraformaldehyde at proestrus or metestrus/diestrus (determined via vaginal cytology). Brains were sectioned at 14 µm and collected in series of five on SuperFrost Plus slides (Roth) and stored at −80 °C. The transsynaptic tracer BL was immunohistologically detected using goat anti-wheat germ agglutinin (1:1000, Vector Laboratories) and has been described previously[31]. A Tyramide Signal Amplification Plus Biotin kit was used for signal amplification. Briefly, slides were washed 3× for 5 min in PBS, incubated in ice-cold methanol with 0.3% $H_2O_2$ for 30 min, washed 3× for 5 min in TNT (0.1 M Tris, 0.15 M NaCl, 0.05% Tween 20), incubated for 10 min in 0.5% Triton X-100 in PBS, washed 3× for 5 min in TNT, blocked with TNB for 30 min, and incubated with anti-wheat germ agglutinin (1:1000) in TNB overnight at 4 °C in a humidified chamber. The next morning, slides were brought to RT for 2 h, washed 3× for 5 min in TNT and incubated with biotinylated anti-goat IgG (1:500) in TNB for 1 h at RT. Slides were then washed 3× for 5 min in TNT and incubated with streptavidin-conjugated horseradish peroxidase (1:100) in TNB for 30 min at RT. Slides were then washed 3× for 5 min in TNT and incubated for 10 min in biotin plus amplification reagent (1:50) in 1× plus amplification diluent at RT. Slides were then washed 3× for 5 min in TNT followed by incubation in streptavidin-conjugated Cy5 (1:500) in TNB for

30 min at RT. Slides were then washed 3× for 5 min with TNT. The slides were incubated overnight at 4 °C followed by 2 h at RT with rabbit anti-nNOS (1:300) in 0.1 M PBS containing 0.5% λ-carrageenan (Sigma) and 0.02% sodium azide. The sections were then treated with Cy3-conjugated donkey anti-rabbit (1:500) in PBS containing 0.5% λ-carrageenan (Sigma) and 0.02% sodium azide in PBS for 1 h at RT. Nuclei were stained with bisbenzimide solution (1:2000 in 0.1 M PBS, 5 min at RT) and coverslipped with Fluoromount-G (Southern Biotech). Images were taken using either a Zeiss Axioskop or a Zeiss Axio Scan Z1 epifluorescence microscope.

**Immunohistochemical detection of mCherry**. To immunohistochemically detect mCherry, slides were washed 3× for 5 min in PBS, incubated in 0.3% Triton X-100, 5% donkey serum, and 0.02% sodium azide in PBS for 1 h at RT followed by anti-ds-red (1:1000, recognizes mCherry) in PBS containing 0.5% λ-carrageenan (Sigma) and 0.02% sodium azide overnight at 4 °C. The next day slides were washed 3× for 5 min with PBS containing 0.5% Tween 20 (PBS+Tween), incubated with Cy3-conjugated donkey anti-rabbit in PBS containing 0.5% λ-carrageenan (Sigma) and 0.02% sodium azide and washed 3× for 5 min in PBS +Tween. Nuclei were stained with bisbenzimide solution (1:2000 in 0.1 M PBS, 5 min at RT) and coverslipped with Fluoromount-G (Southern Biotech). Images were taken using either a Zeiss Axioskop or a Zeiss Axio Scan Z1 epifluorescence microscope.

**Quantification and statistical analysis**. Kisspeptin-immunoreactive (-ir) cell bodies were counted manually and bilaterally in three to four adjacent brain sections (with an interval of 120 μm between them) delineating the RP3V (ante-roventral periventricular area+Periventricular preoptic zone) using a Zeiss Axioskop microscope (×40 objective). Cell counts are expressed as mean number per section for each experimental condition. Analysis of kisspeptin-ir density in the ARC was performed as previously described[69]. To analyze kisspeptin/c-Fos double labeling, three to four sections were selected from the RP3V and the total number of kisspeptin and kisspeptin/c-Fos co-labeled cells were counted in order to obtain the percentage of kisspeptin cells expressing c-Fos immunoreactivity per section.

nNOS-ir and BL+ nNOS-ir cell bodies were counted unilaterally in eight to ten sections containing the VMHvl (Bregma −1.34 to −1.94 according to ref. [63]. Cell counts are expressed as mean number per section for each experimental condition. Statistical significance was determined using Bonferroni's multiple comparison test.

**Statistics**. Randomization was not used in this study and no statistical methods were used to predetermine sample size. Investigators were blinded to the group allocation during experiments or data analysis. Data were analyzed using the GraphPad Prism 7 software. For all statistical comparisons, we first analyzed the data distribution with the Shapiro−Wilk test for normality. Preference score data were analyzed by comparison to hypothesized mean (H0: mean equal 0) using a non-directional one-sample two-tailed *t* test (Figs. 1b, e, 4a and 6a)[70]. For comparison of paired samples comparing two groups, statistical analysis was performed by using a paired-sample two-tailed *t* test (Figs. 2b, d, f, i, 4c and 6d). Comparison of unpaired samples comparing two groups was then performed using an unpaired-sample two-tailed *t* test (Figs. 1c and 2a, e). For comparison of more than two groups, an ANOVA test followed by a Tukey's (Figs. 1e and 6b, c) or Bonferroni's (Fig. 5) two-tailed multiple comparisons test was used. Comparison of more than two data sets violating the normal distribution, the Kruskal−Wallis ANOVA two-tailed test followed by a Dunn's multiple comparison two-tailed test was used (Figs. 1a and 3a, b). Comparison of two data sets violating the normal distribution, a one-sample Wilcoxon two-tailed test was used (Figs. 2c and 4b).

**Resources**. Further details regarding critical resources used for this work including antibodies, virus strains, chemicals kits, and mouse models are listed in Supplementary Table 2.

**Data availability**. Further information and requests for resources and supporting data will be fulfilled by the corresponding authors, J.B. and U.B.

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

## Acknowledgements

We thank Ramona Grünewald for providing excellent technical assistance. This work was supported by grants from the Dutch Science Foundation (Nederlandse Organisatie voor Wetenschappelijk Onderzoek; NWO-VICI 453-08-003) and the Belgian Fonds National de la Recherche Scientifique (FNRS-PDR T.0207.13) to J.B., and the German Science Foundation (Deutsche Forschungsgemeinschaft) through grants SPP1392 and BO1743/6 to U.B. V.H. was also supported by two short-term missions from COST action BM1105, "GnRH deficiency: Elucidation of the neuroendocrine control of human reproduction". J.B. is a research director at the FNRS.

## Author contributions

J.B. and U.B. conceived the study. V.H., O.B., M.C., E.D., M.A. and R.P. conducted the experiments and analyzed data. C.M., A.H., W.H.C. and V.P. provided important tools and reagents. J.B., V.H., O.B., M.C. and U.B. wrote the manuscript.

## Additional information

**Competing interests:** The authors declare no competing financial interests

