## [Peer Review File · Nature Communications]

Reviewers' comments:

Reviewer #1 (Remarks to the Author):

This manuscript by Hellier and coworkers examines the role of kisspeptin neurons in the RP3V in coordination of sex behaviors and ovulation in females. They use a combination of behavioral, molecular, genetic and physiologic approaches to demonstrate that a RP3V to GnRH circuit mediates mate preference whereas an RP3V to VMH/nNOS circuit may mediate lordosis behavior. This is a very interesting set of studies and for the most part the data are clear.

Summary

Sex behavior is needed for survival of more than mammalian species.

GPR54 is an old term; it would be better to switch to the official gene name (Kiss1r) or to at least define GPR54 as Kiss1r.

Introduction

Kiss neurons of the RP3V are one mediator of sex steroid feedback but may not be the only ones (page 4)

Copulation is needed for unassisted fertilization but doesn't ensure it will happen.

Results

A brief introduction of hormonal status, time from treatments and time of day of experiments would be helpful rather than digging through over 20 pages of methods (Table 1 does not include time of day or time from treatment that behavioral tests were done.)

Fonts are too small to be seen in several figures without zooming in to more than 200%.

Activation is used as synonymous with fos expression but fos is activated by growth factors and signaling pathways. Nature 455: 1198 (page 6-adding line numbers would be helpful)

Statistical analyses for many of the studies are inappropriate. The t-test was designed to compare if "two" data sets are different from each other. Zero is no data and one cannot compare something to nothing. Further, there are more than two data sets in most of these studies. The designs are two (or three) way designs and require analysis of variance and post hoc testing. The differences appear robust and using the appropriate stats is not only more rigorous, it will reduce reader questioning of the results because of use of less correct methods. This is true for Fig 1 A B and E, Figs 2, 4, 6, and 7.

For data showing number of kisspeptin-fos double-labeled cells, please provide how many total kisspeptin cells were examined. Did VNOx or Zn affect kisspeptin expression in these cells?

Please put data that are interesting to compare (e.g., fig 1B and E, all LQ data in Fig 3) on the same scale to facilitate comparison.

Individual values should be shown where feasible throughout the figures

The sentence ending "...and drive ovulation in the mammalian brain" on page 7 is an overstatement as none of these studies examined ovulation and not all of them even examined the preovulatory LH surge. These cells are important but one of the coauthors has shown that mice can reproduce when they are ablated suggesting other circuits may exist (Mayer Nat Neurosci)

Page 8 and methods please confirm the GnRH dose as 250 nanograms/kg produces a physiological pituitary response. If not a typo please provide a rationale for using such a high dose

Figure S3 the double labeling wasn't convincing until blown up to 400%. Please either choose a clearer example or enlarge.

Is the third group for this Kruskal Wallis test discussed for fig 3 D the kiss-/- in figure 3C? It would help to have all the data being compared in the same figure or at minimum clarified. What was the post hoc (Dunn's?)

Cre -/- animals should be included as a control for 3I. Is this frequency and interval of stimulation based on endogenous firing in these cells?

This stimulation pattern needs to be clarified in the methods in which it appears to state that 10 trains of stimulation lasting 15 s were given every 60 s-this doesn't seem possible.

Does the apparent increase in synaptic communication mentioned on page 10 reflect the increase in nNos expression? Given the number of days from injection of the tracer to study, it is surprising that there would be a cycle dependent change in BL labeling of nNos neurons, and Figure 6B suggests this is not the case. (page 12)

Page 11 bottom incorrect ref to fig 6 A instead of 7A

Discussion

Pheromones act through kisspeptin neurons to induce opposite-sex mate preferences section

Behaviors are sexually differentiated, not dimorphic p12.

Several statements are made without citations (e.g., "Many

13

mammalian species use olfactory cues to identify and to attract potential mates and secrete a

wide variety of odors into the environment which can lead to changes in behavior and/or reproductive physiology in the recipient.")

This section wanders a bit and could benefit from breaking into a couple paragraphs with distinct points.

Kisspeptin-GnRH signaling section

If lordosis is normal in GPR54 KO mice (Kauffman 2007), what is the presumed mode of RP3V kisspeptin neuron communication with nNOS neurons in the VMH?

The sentence beginning with "Finally" is a run-on that is hard to follow.

Kisspeptin, a new therapeutical drug in treating low sexual desire? Section

This section seems to go a bit far. The statement that low sexual desire is very common is perhaps too strong given it is based on an industry-sponsored supplement to a journal.

Methods

To be frank, these are very difficult to follow. This work is clearly a collaborative effort but there needs to be some attempt to consolidate methods (e.g., OVX+E is mentioned at start but implant construction isn't until several pages later) or at least present the same information (for example lots of immuno but very different presentations. There are several complex experimental timelines and one really has to look to find when something was done relative to something else. A supplementary figure could illustrate this very easily, perhaps a design timeline for each main figure.

Time between behavioral tests is clear for some studies but not others (e.g., after photostimulation)

Please make sure all doses are given as mass or moles per mouse body weight. Volume is fine to include but readers should not have to calculate doses (e.g., SNAP treatment section)

Please provide the rationale for why the same dose of kisspeptin 10 was given ICV and IP.

Page 42 citation needed for antifreeze solution

Page 43, was 30% H₂O₂ really used?

Reviewer #2 (Remarks to the Author):

The authors used a wide variety of methods (transgenic mice, viral vectors, optogenetics, electrophysiology, lesions, tract tracing, immunohistochemistry, behavioral testing) to assess the role of hypothalamic kisspeptin neurons in regulating the preovulatory GnRH

surge and estrous behaviors. The work is generally well presented and the findings are interesting. The data suggest that sexually dimorphic kisspeptin neurons in the RP3V act through two parallel pathways, one involving projections to GnRH neurons and another involving projections to nitric oxide-synthesizing neurons in the vVMH, to coordinate ovulation with the period of behavioral estrus. The experiments also document a key role for olfactory stimuli processed by the vomeronasal pathway. Thus the work as a whole demonstrates kisspeptinergic neural circuits that are essential for reproductive success in female mice. There is a concern about the statistical approach (see item 1 in Major Comments). Both the major and minor concerns detailed below should be addressable in a revised manuscript.

Major comments:

1. The Methods section states that all parametric statistical tests used the paired Student's t-test. This test is used to compare the same subjects under two different conditions. This is clearly not the case for many of the experiments reported in the manuscript (e.g., in Fig 1C, it is not correct to compare animals of 2 different genotypes using a paired t-test; this is just the first example.). In some cases, there are more than two groups being compared (see Figure 1A and many others), which cannot be done appropriately with a Student t-test until ANOVA (or its non-parametric equivalent) is done first. Also, with respect to Figure 1A and 1C, the definition of the symbols in the figure legend does not make sense. It is also unclear to this reviewer how the preference scores were compared statistically to "0", or no preference. Were the ovx animals first tested with no hormone treatment, found to have no preference, and then the scores compared to when the same animals were treated with estradiol and progesterone? Therefore, the statistical analysis needs to be reconsidered and described in more detail.
2. The authors seemed surprised to find that GnRH::Cre; DicerloxP/loxP females showed normal levels of lordosis behavior. In fact, this was to be expected as Marie Gibson and colleagues demonstrated 20 years ago that hypogonadal (hpg) mice, which carry a deletion in the GnRH gene that results in no GnRH peptide being produced, showed that GnRH is not essential for reproductive behaviors in either males or females. They should cite this work and perhaps modify the language a bit.
3. The final paragraph of the discussion, regarding the possibility that kisspeptin may be useful for treating low sexual desire, is highly speculative and should be toned down at the very least, or preferably deleted. The studies reported here measured lordosis behavior and mate preference (choice of intact male or estrous female), neither of which is a good proxy for low sexual desire in women. Other behavioral tests (e.g., paced mating, conditioned place preference) would be needed to support this speculation.

Minor comments:

1. Results, p. 6: In describing Figure 1B, the authors state that Kp-10 "triggered a male-directed preference" in control littermates. But the preference scores in saline and Kp-10 infused animals look identical.
2. Results, p. 7: As written, the following sentence states that the brain is the site of ovulation: "RP3V kisspeptin neurons directly innervate GnRH neurons (Liu et al., JN 31

2011. Yip et al., 2015) and drive ovulation in the mammalian brain." Please rewrite the sentence.

3. Results, p. 9: In describing Figure 3I, the authors state: "Photostimulation of RP3V kisspeptin neurons at 10 Hz for < 15s per male mount also was successful in triggering lordosis behavior in Cre+ female mice (Student's t test; P = 0.049; Figure 3I)." In fact, LQs in the unstimulated animals were already around 35, and the stimulation increased this somewhat (to about 50). So the sentence should be modified to indicate that stimulation enhanced or facilitated lordosis rather than triggering the behavior.

4. Please add "n" to the bars in Figures 6H and 6I as was done in other figures.

5. Results, pp. 11 and 12: For those who are not familiar with the pharmacology, it would be helpful to note what the BAY compound does. This information is in the Methods, but if added to the Results would help some readers.

6. Discussion, p. 16: When the authors note that progesterin receptors are expressed in the vVMH and that their expression is highly dependent on estrogens, they cite their own work published in 2010. It would be more appropriate to cite the earlier work of investigators such as Blaustein and McEwen who first demonstrated this decades earlier.

7. It is probably more correct to use the term "progesterin receptors" rather than "progesterone receptors".

8. There are a few instances (e.g., middle of p. 11) in which the noun form of the word "estrus/diestrus" is used when the adjective form "estrous/diestrous) should be used.

Reviewer #3 (Remarks to the Author):

see attached

The reported data in this manuscript make a significant contribution to the field by detailing for the first time requisite and distinct mechanisms that underlie sexual preference and sexual behavior (lordosis) in females. These data build on previous work by this lab. A notable contribution is finding that both sexual preference and lordosis hinge on the pheromonal processing vomeronasal organ (VNO) and kisspeptin neurons in the AVPV, but then the story diverges, with GnRH neurons critically mediating mate choice but not lordosis and kisspeptin-recipient nNos⁺ neurons in the VMHvl mediating lordosis (and perhaps not mate choice?). The investigators use powerful cre-lox and other genetic tools combined with optogenetics and viral delivery to get at questions that have longed plagued the field. With the VNO, kisspeptin neurons and the VMHvl all implicated in determining mate choice and expression of female sex behavior, this work begins to unravel the distinct neural mechanisms that control distinct parts of a constellation of coordinated behaviors that we call “female sex behavior.” The authors should also be commended for the thorough disclosure of the reagents used in their studied. While this is judged as significant work with broad impact, presentation of the work needs improvement, as outlined below.

Major concerns:

1. The title is misleading and should be revised. No data is presented on whether the circuits studied are sexually differentiated (since only females are studied) nor whether the circuits studied synchronize ovulation with sex behavior. While this seems likely based on circumstantial evidence, no data presented in the manuscript shows this. In fact, for the most part, experimental females are ovariectomized.
2. The story would be easier to follow if it was reorganized. For example, the initial expts tell the same story for both mate choice and lordosis—depends on VNO and kisspeptin, but then the story diverges when GnRH disrupts mate choice but not lordosis. These data in particular should be presented in the same figures (e.g., combine fig 2 with 5) so this divergence is underscored.
3. The use of many diverse models is commendable. However, readers would benefit from a brief description of the model and its validation in the results section as data are presented on each model. How well each model is validated is a particular concern, particularly in regard to the specificity of the targeted deletions. The validation of these models should be addressed.
4. When lordosis behavior is recovered in a given model by treatment (e.g., Kp-10), it would be helpful to tell readers in the Results sections the status of the females (e.g., Ovex and E2-primed but no progesterone). It becomes quite tedious to have to keep referring to the supplemental Table.
5. An important control when using conditional knockouts (KOs) is to show that the effect depends on the *combination* of the two (cre + lox). What are the results when AAV vectors containing loxp constructs are injected in cre negative mice? Pertinent to Fig 3I.
 - a. It is important that the authors be clear that the electrophysiological data comes from slices and not in vivo (page 9, middle paragraph).
6. Figures: The figures are not very reader-friendly and could be improved significantly by adding more headers and labels to make clear what model the data comes from.
 - a. For example, data for A, B and C of fig 1 each come from different models. Somehow this needs to be made clearer in the figure itself. One of the important distinctions between data for B vs C is that one is a global kisspeptin KO, presumably embryonic, whereas C comes from a model where it is a targeted kisspeptin KO in the AVPV (sparing

the arcuate) in adulthood. Reference to relevant supplemental data would also be helpful if it were added to the caption

- b. Labels (axis and legends) in general need to be larger.
 - c. For all graphs of preference score data, it would be helpful to include up and down arrows (from zero) and the two labels (intact males and estrous females) throughout.
 - d. Figure 6H and I:
 - i. The Ns are not indicated in the bars for the presented graphs as in every other graph and should be included. The Ns are low and likely underpowered. This issue should be addressed in the discussion and interpretation of these data.
 - ii. A simpler and more straight forward way to present these data are # of nNOS+, # of BL+ and % nNOS+ BL neurons in a single graph and eliminate "I".
7. Statistics:
- a. Students T tests seems to be over-used with no apparent correction (e.g., Bonferroni) for repeated T tests on the same data set. There are times when an ANOVAs would clearly be more appropriate and the probability of a type I error can be held at alpha $p = 0.05$ by selecting the appropriate post hoc tests. A specific example is the data presented in Fig 1E (assessing the effects of treatment versus cre could be analyzed via a 2-way Anova with Tukey or Bonferroni post hoc comparisons).
 - b. It is inappropriate to directly compare two groups that differ on two variables. All such reported significances should be deleted.
 - c. Lordosis behavior is sometimes analyzed using student T tests and other times Mann Whitney U. While a general explanation is offered in the supplement, this should be directly addressed in the main body of the paper. In particular, lordosis data are sometimes deemed normally distributed and sometimes not. This should be addressed.
 - d. The statistical test used is generally reported in both the caption and the Results section but this information is sometimes missing in one place or another. Whatever the authors decide to do, it should be done consistently throughout the paper.
8. Explain the nature of the control group (Fig 1 and 4), when data on it is presented and the reason for the comparatively high (28) N for this particular group.
9. Relevant to the data presented in Fig 7, it would be more compelling if the authors showed that the nNos treatment (C) in kiss KO mice was specific to lordosis with *no* effect on mate preference, complimenting the effect of GnRH that was specific to preference but with no effect on lordosis. Other data that could easily have been reported to address this issue is examining both mate choice and lordosis behavior in nNOS KO to determine whether *only* lordosis but not mate choice was affected. These crucial data are lacking and would heavily influence the conclusions drawn. This is a particularly important issue, given that VMHvl neurons are also implicated in the control of mate choice in females (see supplemental data, <https://www.ncbi.nlm.nih.gov/pubmed/24739975>).

Minor concerns:

1. First two sentences of intro are not informative and could be deleted.
2. Title sentence for Fig 2 caption needs to be reworded since the data shown addresses the dependence of preference on GnRH and *not* kisspeptin. The same concern is pertinent to Fig 7 caption.

3. Title sentence for Fig 6 caption needs reworking. Consider: "Both nNos expression and kisspeptin recipient neurons are increased in the VMHvl in proestrus."
4. The data shown in figure 6 do not actually show direct synaptic connections between BL neurons in the VMH and kisspeptin neurons from the AVPV but their data make this likely. I encourage the authors to be more circumspect in how they describe these data.

Responses to reviewers' comments:

Manuscript NCOMMS-17-14543: "A sexually dimorphic neural circuit synchronizes female sexual behavior with ovulation".

We thank the reviewers for their thoughtful and constructive comments on our manuscript. Each of these are addressed as described below, either by the addition of figures, changes to the manuscript, or explanatory responses. All changes in the revised manuscript are highlighted in blue.

Reviewer #1:

General comments

1. Sex behavior is needed for survival of more than mammalian species.

Response: This sentence has been deleted in response to a comment from Reviewer 3.

2. GPR54 is an old term; it would be better to switch to the official gene name (Kiss1r) or to at least define GPR54 as Kiss1r.

Response: We have replaced GPR54 with Kiss1R as requested.

3. Kiss neurons of the RP3V are one mediator of sex steroid feedback but may not be the only ones (page 4)

Response: We have changed this to "Instead, gonadal sex steroid feedback is indirectly relayed to GnRH neurons. One critical neuronal population relaying sex steroid feedback to GnRH neurons lies in the rostral periventricular area of the third ventricle (RP3V) of the hypothalamus and produces the neuropeptide kisspeptin".

4. Copulation is needed for unassisted fertilization but doesn't ensure it will happen.

Response: This sentence has been deleted in response to a comment from Reviewer 3.

5. A brief introduction of hormonal status, time from treatments and time of day of experiments would be helpful rather than digging through over 20 pages of methods (Table 1 does not include time of day or time from treatment that behavioral tests were done.)

Response: We have added a paragraph explaining the hormonal treatments, the time of day of testing and the natural variability in female sexual behavior observed in the control females right at the beginning of the Behavioral tests section in the Methods and refer the reader to this paragraph in the first paragraph of the Results section. We also have added the individual hormonal treatments to the main text whenever possible (i.e. OVX+E or OVX+E+P). We have in addition specified in the table that progesterone was administered 3h prior to behavioral experiments.

6. Fonts are too small to be seen in several figures without zooming in to more than 200%.

Response: Fonts have been enlarged in all the figures as requested

7. Activation is used as synonymous with fos expression but fos is activated by growth factors and signaling pathways. Nature 455: 1198 (page 6-adding line numbers would be helpful)

Response: We have changed this to clarify that we are referring to c-Fos expression: "...male odor-triggered kisspeptin neuron activation in these animals by using c-Fos as a marker" and "...demonstrating that pheromonal input triggers c-Fos expression in kisspeptin neurons via the vomeronasal pathway."

8. Statistical analyses for many of the studies are inappropriate. The t-test was designed to compare if "two" data sets are different from each other. Zero is no data and one cannot compare something to nothing. Further, there are more than two data sets in most of these studies. The designs are two (or three) way designs and require analysis of variance and post hoc testing. The differences appear robust and using the appropriate stats is not only more rigorous, it will reduce reader questioning of the results because of use of less correct methods. This is true for Fig 1 A B and E, Figs 2, 4, 6, and 7.

Response: Figures 1A and 4 (now Fig 3). Thank you for pointing this out, it turned out that we had indeed made a mistake by applying a Student's test here. Since the distribution of the data was not normal, a Kruskal Wallis test followed by a Dunn test for multiple comparisons was used. The P values and number of stars for each graph have been changed in the main text and in the figures.

Figures 1B/-, 1E and 2 (now Fig 4a). Since we wanted to analyze the ability of females to choose between a stimulus male and a stimulus female, a Student's test allowed us to compare the scores to a reference value which is 0 in our case (but which does not represent a real value here). This method has been commonly used to analyze preference data (e.g. Boillat et al, Current Biology 2015; 25: 251-255; LeGates et al, Nature 2012; 491: 594-598).

Figure 6 (now Fig 5): we now used an ANOVA analysis followed by Bonferroni's multiple comparison test.

Figure 7 (now Figure 6 in the new version): We now used an ANOVA analysis as suggested for the data presented in Figures 6b and 6c (injection of SNAP or either Kp-10 or GnRH on lordosis behavior in nNOS^{-/-} mice). P values have been changed in accordance to the statistical analyses in the main text. For Fig 6d, we kept the student t-test because we only compared two data sets.

9. For data showing number of kisspeptin-fos double-labeled cells, please provide how many total kisspeptin cells were examined. Did VNOx or Zn affect kisspeptin expression in these cells?

Response: The number of kisspeptin neurons was not significantly different between Sham (80.7 ± 3.89), VNOx (72.7 ± 7.74), Zinc sulfate application (64.8 ± 5.899) or the combination of both VNO ablation and ZnSO₄ treatment (66.7 ± 7.15). We have added a sentence to the results section.

10. Please put data that are interesting to compare (e.g., fig 1B and E, all LQ data in Fig 3) on the same scale to facilitate comparison.

Response: We have changed this as requested in figure 1 (regarding preference scores) and figure 2 (formerly Fig 3) (regarding LQ data).

11. Individual values should be shown where feasible throughout the figures

Response: We have changed this as requested for all the figures

12. The sentence ending "...and drive ovulation in the mammalian brain" on page 7 is an overstatement as none of these studies examined ovulation and not all of them even examined the preovulatory LH surge. These cells are important but one of the coauthors has shown that mice can reproduce when they are ablated suggesting other circuits may exist (Mayer Nat Neurosci)

Response: We have changed this to "...and have been implicated in generating the preovulatory LH surge"

13. Page 8 and methods please confirm the GnRH dose as 250 nanograms/kg produces a physiological pituitary response. If not a typo please provide a rationale for using such a high dose

Response: This was actually a mistake. We injected 5 µg of GnRH per mouse (± 25g) which is a dose of 25 nanograms/kg. This dose was based on previous studies in our lab (Keller et al, EJN 2006) as well as on papers by Pfaff, Science 1973, Mackay-Sim & Rose, Neuroendocrinology 1986, and Saito & Moltz, Physiology and Behavior 1986.

14. Figure S3 the double labeling wasn't convincing until blown up to 400%. Please either choose a clearer example or enlarge.

Response: We have tried to take better pictures but these new ones were no better than the original ones. Unfortunately, it remains very difficult to convincingly show double-labeling immunohistochemistry. When focusing too much on kisspeptin, the Fos gets blurry, and vice versa.

15. Is the third group for this Kruskal Wallis test discussed for fig 3 D the kiss^{-/-} in figure 3C? It would help to have all the data being compared in the same figure or at minimum clarified. What was the post hoc (Dunn's?)

Response: Indeed, they are the same data. We wanted to avoid duplications, i.e. showing the same result twice, i.e. that an injection with KP-10 stimulates lordosis behavior in Kiss^{+/+} females as was observed in WT C57BL6/j mice in Fig 2B (formerly 3B). We have now changed Fig 2D (formerly 3D) to show that a single KP-10 injection stimulated lordosis behavior in Kiss^{-/-} females. Finally, regarding the statistical analyses, a Student's t test has been used since the data were distributed normally, which revealed a significant increase in the expression of lordosis behavior ($t=4.42$, $p<0.001$).

17. Cre^{-/-} animals should be included as a control for 3I. Is this frequency and interval of stimulation based on endogenous firing in these cells?

Response: 10 Hz was chosen as electrical stimulation because this frequency has been shown to elicit kisspeptin release from AVPV kisspeptin neurons (Liu et al., 2011 JNeuroSci). Optogenetic stimulation at 10 Hz in ARC kisspeptin neurons has also been demonstrated to trigger kisspeptin release (Han et al., 2015 PNAS). We have now added data on Cre⁻ animals injected with the Chr2 virus. No enhancing effects on lordosis behavior were observed upon photostimulation.

18. This stimulation pattern needs to be clarified in the methods in which it appears to state that 10 trains of stimulation lasting 15 s were given every 60 s-this doesn't seem possible.

Response: *We have now clarified that the optogenetic stimulation occurred at 10Hz for periods of 1-15s closely following the protocol used in vivo, i.e. whenever the male approached the female. As noted above in the response to point 17, 10Hz is the minimal frequency required to generate kisspeptin release from RP3V kisspeptin neurons and this is most effective when given in an intermittent bursting manner. Hence, in our experimental protocol we intermittently activated RP3V kisspeptin neurons in a behaviourally-relevant context for the female.*

19. Does the apparent increase in synaptic communication mentioned on page 10 reflect the increase in nNos expression? Given the number of days from injection of the tracer to study, it is surprising that there would be a cycle dependent change in BL labeling of nNos neurons, and Figure 6B suggests this is not the case. (page 12)

Response: *These data sets have also now been re-analysed using Bonferonni's multiple comparisons test rather than one-tailed Student's t-test, as requested by reviewer 3. As a result, the difference between # of BL+ neurons (proestus) and # of BL+ neurons (met/diestrus) is non-significant. The language in the manuscript has been modified to reflect this and this issue has been addressed in the discussion.*

20. Page 11 bottom incorrect ref to fig 6 A instead of 7A

Response: *Changed as requested but note that figure numbers have been changed.*

Discussion

21. Pheromones act through kisspeptin neurons to induce opposite-sex mate preferences section Behaviors are sexually differentiated, not dimorphic p12.

Response: *Changed as requested. "Mate preferences are sexually differentiated".*

22. Several statements are made without citations (e.g., "Many 13 mammalian species use olfactory cues to identify and to attract potential mates and secrete a wide variety of odors into the environment which can lead to changes in behavior and/or reproductive physiology in the recipient.")

Response: *We have added the references (Brown, 1979; Bakker, 2003).*

23. This section wanders a bit and could benefit from breaking into a couple paragraphs with distinct points.

Response: *We have shortened this section considerably in the revised version of the manuscript.*

24. If lordosis is normal in GPR54 KO mice (Kauffman 2007), what is the presumed mode of RP3V kisspeptin neuron communication with nNOS neurons in the VMH?

Response: *We have now suggested a potential mechanism underlying kisspeptin signaling in the absence of GPR54. "Several studies (Lyubimov et al., 2010; Oishi et al., 2011; Elhabazi et al., 2013) have demonstrated that kisspeptin can activate neuropeptide FF receptors (NPFFR1 and NPFFR2). Herbison and colleagues (Liu and Herbison, 2015)*

have demonstrated that kisspeptin can modulate arcuate neuron excitability at least partially via NPF receptors independently of Kiss1R. As NPFFR2 and (to a lesser extent) NPFFR1 have been detected in the VMH of mice (Gouardères, Puget and Zajac, 2004), it is tempting to speculate that these receptors may play a key role in the facilitation of lordosis behavior. Future studies using NPF receptor mutant mice may provide new evidence of the involvement of these receptors in the kisspeptidergic modulation of reproductive behavior. Taken together, our... ”.

25. The sentence beginning with “Finally” is a run-on that is hard to follow.

Response: This has now been shortered to “Finally, it is quite likely that specialized subpopulations of kisspeptin neurons exist within the RP3V since transynaptic tracing has indicated that the majority of kisspeptin neurons within the RP3V does not communicate with GnRH neurons.”

26. Kisspeptin, a new therapeutical drug in treating low sexual desire? Section
This section seems to go a bit far. The statement that low sexual desire is very common is perhaps too strong given it is based on an industry-sponsored supplement to a journal.

Response: We have now truncated this section in accordance with reviewer comments and retitled the section “The kisspeptidergic system as a novel drug target for low sexual desire”

Methods

27. To be frank, these are very difficult to follow. This work is clearly a collaborative effort but there needs to be some attempt to consolidate methods (e.g., OVX+E is mentioned at start but implant construction isn't until several pages later) or at least present the same information (for example lots of immuno but very different presentations. There are several complex experimental timelines and one really has to look to find when something was done relative to something else. A supplementary figure could illustrate this very easily, perhaps a design timeline for each main figure.

Response: We have reorganized the Methods. We hope that it is clearer now what we have done in each experiment. We now also mention the different hormone treatments (OVX + E versus OVX + E +P) in the Results section.

28. Time between behavioral tests is clear for some studies but not others (e.g., after photostimulation)

Response: Generally, females were tested for lordosis behavior once a week. We have added that the 5th test was performed one week later to the description of the lordosis tests with photostimulation.

29. Please make sure all doses are given as mass or moles per mouse body weight. Volume is fine to include but readers should not have to calculate doses (e.g., SNAP treatment section)

Response: This information has been changed as requested for the SNAP treatment which was administered at a dose of 8 mg/kg

30. Please provide the rationale for why the same dose of kisspeptin 10 was given ICV and IP.

Response: In fact, we did not use the same dose for icv and sc as it turned out. We used 100 nM as dose, but the final dose delivered depends on the volume injected (100 µl sc versus 2 µl icv). It should be noted that we made a mistake in recalculating our dose of KP-10 from nanomolar to mg/kg. So we actually injected 0.52 µg/kg sc and 10.4 ng/kg icv. We have corrected this throughout the ms.

31. Page 42 citation needed for antifreeze solution

Response: Citation has been added.

32. Page 43, was 30% H₂O₂ really used?

Response: Thank you for picking this up, this was a typo. “3% hydrogen peroxide” has been changed to “0.3% hydrogen peroxide”

Reviewer #2:

Major comments:

33. The Methods section states that all parametric statistical tests used the paired Student’s t-test. This test is used to compare the same subjects under two different conditions. This is clearly not the case for many of the experiments reported in the manuscript (e.g., in Fig 1C, it is not correct to compare animals of 2 different genotypes using a paired t-test; this is just the first example.). In some cases, there are more than two groups being compared (see Figure 1A and many others), which cannot be done appropriately with a Student t-test until ANOVA (or its non-parametric equivalent) is done first. Also, with respect to Figure 1A and 1C, the definition of the symbols in the figure legend does not make sense. It is also unclear to this reviewer how the preference scores were compared statistically to “0”, or no preference. Were the ovx animals first tested with no hormone treatment, found to have no preference, and then the scores compared to when the same animals were treated with estradiol and progesterone? Therefore, the statistical analysis needs to be reconsidered and described in more detail.

Response: We agree that the description of the statistical analyses lacked important details. It should be noted that we never used any paired t tests to compare animals of different genotypes. This was misstated and has been corrected (regarding Fig 1C). We have now added a more detailed description of the statistical analyses used throughout the manuscript (in the results and method sections). We have also changed some analyses because some of the data was not normally distributed (for instance regarding the data presented in Fig 1A and 4, now Fig 3). We have now used a Kruskal Wallis test following a Dunn multiple comparison test.

Regarding the statistical analyses used for the mate preference scores, the “0” means that the animal showed no preference (since a preference score was calculated by distracting the time spent sniffing the female side from the time spent sniffing the male side, divided by the total amount of time spent sniffing both sides).

In the graphs presenting the data of mate preferences, “0” represents the “no preference” meaning a preference score of 0, females are considered to have no preference. Since the aim of these experiments was to analyze the effect of a mutation or treatment on mate preference, each condition was compared to the neutral 0 value with the help of a Student’s t test, with one exception in fig 1E where we were primarily interested in the difference between Cre- and Cre+ females.

34. The authors seemed surprised to find that GnRH::Cre; DicerloxP/loxP females showed normal levels of lordosis behavior. In fact, this was to be expected as Marie Gibson and colleagues demonstrated 20 years ago that hypogonadal (hpg) mice, which carry a deletion in the GnRH gene that results in no GnRH peptide being produced, showed that GnRH is not essential for reproductive behaviors in either males or females. They should cite this work and perhaps modify the language a bit.

Response: The language has been modified as requested and the previous studies in hpg mice cited. "These data are consistent with previous studies in mice harboring a deletion in the GnRH gene (and therefore fail to synthesize GnRH), which also appear (after OVX and priming with estradiol and progesterone) to have no deficits in sexual behavior (Ward and Charlton, 1981)."

35. The final paragraph of the discussion, regarding the possibility that kisspeptin may be useful for treating low sexual desire, is highly speculative and should be toned down at the very least, or preferably deleted. The studies reported here measured lordosis behavior and mate preference (choice of intact male or estrous female), neither of which is a good proxy for low sexual desire in women. Other behavioral tests (e.g., paced mating, conditioned place preference) would be needed to support this speculation.

Response: We have now truncated this section in accordance with reviewer comments and retitled the section "The kisspeptidergic system as a novel drug target for low sexual desire"

Minor comments:

36. Results, p. 6: In describing Figure 1B, the authors state that Kp-10 "triggered a male-directed preference" in control littermates. But the preference scores in saline and Kp-10 infused animals look identical.

Response: It is true that KP did not further increase the preference in control animals. We have therefore deleted this particular statement.

37. Results, p. 7: As written, the following sentence states that the brain is the site of ovulation: "RP3V kisspeptin neurons directly innervate GnRH neurons (Liu et al., JN 31 2011. Yip et al., 2015) and drive ovulation in the mammalian brain." Please rewrite the sentence.

Response: This has been changed in accordance with a comment from Reviewer 1. "RP3V kisspeptin neurons directly innervate GnRH neurons (Liu et al., JN 31 2011. Yip et al., 2015) and have been implicated in generating the preovulatory LH surge".

38. Results, p. 9: In describing Figure 3I, the authors state: "Photostimulation of RP3V kisspeptin neurons at 10 Hz for < 15s per male mount also was successful in triggering lordosis behavior in Cre+ female mice (Student's t test; P = 0.049; Figure 3I)." In fact, LQs in the unstimulated animals were already around 35, and the stimulation increased this somewhat (to about 50). So the sentence should be modified to indicate that stimulation enhanced or facilitated lordosis rather than triggering the behavior.

Response: Changed as requested "successful in enhancing lordosis".

39. Please add “n” to the bars in Figures 6H and 6I as was done in other figures.

Response: This figure has been modified and now displays the individual data points.

40. Results, pp. 11 and 12: For those who are not familiar with the pharmacology, it would be helpful to note what the BAY compound does. This information is in the Methods, but if added to the Results would help some readers.

Response: Changed as requested “...treated with a cocktail of the nitric oxide donor SNAP and the guanylate cyclase agonist BAY 41-2272”.

41. Discussion, p. 16: When the authors note that progesterin receptors are expressed in the v1VMH and that their expression is highly dependent on estrogens, they cite their own work published in 2010. It would be more appropriate to cite the earlier work of investigators such as Blaustein and McEwen who first demonstrated this decades earlier.

Response: Changed as requested. Of note, a recent study (Chacklaki et al, J Comp Neurology 2017) showed that close to 100% of VMHvl neurons expressing estradiol receptors are nNOS expressing neurons further suggesting an important role for this particular population in female sexual behavior.

42. It is probably more correct to use the term “progesterin receptors” rather than “progesterone receptors”.

Response: Changed as requested.

43. There are a few instances (e.g., middle of p. 11) in which the noun form of the word “estrus/diestrus” is used when the adjective form “estrous/diestrous” should be used.

Response: Changed to “compared with females in diestrus”.

Reviewer #3:

Major concerns:

44. The title is misleading and should be revised. No data is presented on whether the circuits studied are sexually differentiated (since only females are studied) nor whether the circuits studied synchronize ovulation with sex behavior. While this seems likely based on circumstantial evidence, no data presented in the manuscript shows this. In fact, for the most part, experimental females are ovariectomized.

Response: We agree with this criticism and have changed our title to kisspeptin neurons synchronize female sexual behavior with ovulation.

45. The story would be easier to follow if it was reorganized. For example, the initial expts tell the same story for both mate choice and lordosis—depends on VNO and kisspeptin, but then the story diverges when GnRH disrupts mate choice but not lordosis. These data in particular should be presented in the same figures (e.g., combine fig 2 with 5) so this divergence is underscored.

Response: We agree with the reviewer and we have now combined the section on kisspeptin-GnRH signaling in mate preference with the one on the role of GnRH in lordosis behavior (which will come after the two sections on kisspeptin on mate preferences and lordosis behavior). We have thus combined fig 2 with fig 5 in one figure (now figure 4).

46. The use of many diverse models is commendable. However, readers would benefit from a brief description of the model and its validation in the results section as data are presented on each model. How well each model is validated is a particular concern, particularly in regard to the specificity of the targeted deletions. The validation of these models should be addressed.

Response: Since all mouse models used in the present study have already been published and thus well validated before, we feel that it would be somewhat redundant to provide this information again in the Results section.

47. When lordosis behavior is recovered in a given model by treatment (e.g., Kp-10), it would be helpful to tell readers in the Results sections the status of the females (e.g., Ovex and E2-primed but no progesterone). It becomes quite tedious to have to keep referring to the supplemental Table.

Response: We have now included the status of the females in the Results section as requested.

48. An important control when using conditional knockouts (KOs) is to show that the effect depends on the combination of the two (cre + lox). What are the results when AAV vectors containing loxp constructs are injected in cre negative mice? Pertinent to Fig 3I.

Response: We have now added the data on Cre- females injected with the ChR2 virus and no enhancing effects were observed on lordosis behavior upon photostimulation.

49. It is important that the authors be clear that the electrophysiological data comes from slices and not in vivo (page 9, middle paragraph).

Response: Changed as requested: "...fidelity of 99% (Figure 3H) in brain slice preparations. Photostimulation of RP3V kisspeptin neurons in vivo..."

50. Figures: The figures are not very reader-friendly and could be improved significantly by adding more headers and labels to make clear what model the data comes from.

a. For example, data for A, B and C of fig 1 each come from different models. Somehow this needs to be made clearer in the figure itself. One of the important distinctions between data for B vs C is that one is a global kisspeptin KO, presumably embryonic, whereas C comes from a model where it is a targeted kisspeptin KO in the AVPV (sparing the arcuate) in adulthood. Reference to relevant supplemental data would also be helpful if it were added to the caption

Response: Headers have been added to the figures as requested.

50.b. Labels (axis and legends) in general need to be larger.

Response: Changed as requested for all the figures.

50.c. For all graphs of preference score data, it would be helpful to include up and down arrows (from zero) and the two labels (intact males and estrous females) throughout.

Response: The two labels were missing in figure 2 (now figure 4A) and have been added as requested and arrows have also been added to facilitate the reading of the figures as requested

50.d. Figure 6H and I:

i. The Ns are not indicated in the bars for the presented graphs as in every other graph and should be included. The Ns are low and likely underpowered. This issue should be addressed in the discussion and interpretation of these data.

Response: The graph has been changed and now shows individual data points. These data sets have also now been re-analysed using Bonferonni's multiple comparisons test rather than one-tailed Student's t-test, as requested. As a result, the difference between # of BL+ neurons (proestus) and # of BL+ neurons (met/diestrus) is non-significant. The language in the manuscript has been modified to reflect this.

ii. A simpler and more straight forward way to present these data are # of nNOS+, # of BL+ and % nNOS+ BL neurons in a single graph and eliminate "I".

Response: Changed as requested.

51. Statistics:

51.a. Students T tests seems to be over-used with no apparent correction (e.g., Bonferroni) for repeated T tests on the same data set. There are times when an ANOVAs would clearly be more appropriate and the probability of a type I error can be held at alpha $p = 0.05$ by selecting the appropriate post hoc tests. A specific example is the data presented in Fig 1E (assessing the effects of treatment versus cre could be analyzed via a 2-way Anova with Tukey or Bonferroni post hoc comparisons).

Response: Some of the statistical analyses have been changed either by a 2-way ANOVA (Figure 6, formerly figure 7) or by a Kruskal Wallis test completed by Dunn's comparison test (Figures 1 and 3).

51.b. It is inappropriate to directly compare two groups that differ on two variables. All such reported significances should be deleted.

Response: This is not exactly what we did. The description of the statistics in the methods missed some important details notably about the use of Student's t test. The description of the statistical analyses has been added to the main text, the figures, figure legends and in the methods to avoid further misunderstanding. We also found that some statistical analyses were not correct and we have changed those accordingly.

51.c. Lordosis behavior is sometimes analyzed using student T tests and other times Mann Whitney U. While a general explanation is offered in the supplement, this should be directly addressed in the main body of the paper. In particular, lordosis data are sometimes deemed normally distributed and sometimes not. This should be addressed.

Response: Changed as requested

51.d. The statistical test used is generally reported in both the caption and the Results section but this information is sometimes missing in one place or another. Whatever the authors decide to do, it should be done consistently throughout the paper.

Response: *Changed as requested to make it consistent throughout the paper.*

52. Explain the nature of the control group (Fig 1 and 4), when data on it is presented and the reason for the comparatively high (28) N for this particular group.

Response: *In figures 1 and 3 (formerly fig 4), the control group consisted of all the mice which had not been exposed to either male odors (Figure 1) or a male for mating (Figure 3). Since we included controls for each surgical intervention (VNO removal, zinc sulfate treatment, or both interventions), this led to a total of 28 “sham” animals. Since the statistical analyses showed no significant differences in the number of Fos/Kp colabelled cells between the sham controls, we decided to combine them in one single control group.*

53. Relevant to the data presented in Fig 7, it would be more compelling if the authors showed that the nNos treatment (C) in kiss KO mice was specific to lordosis with no effect on mate preference, complimenting the effect of GnRH that was specific to preference but with no effect on lordosis. Other data that could easily have been reported to address this issue is examining both mate choice and lordosis behavior in nNOS KO to determine whether only lordosis but not mate choice was affected. These crucial data are lacking and would heavily influence the conclusions drawn. This is a particularly important issue, given that VMHvl neurons are also implicated in the control of mate choice in females (see supplemental data, <https://www.ncbi.nlm.nih.gov/pubmed/24739975>).

Response: *We have now added data on mate preferences in nNOS KO females. They are also disrupted, i.e. they failed to show a male-directed preference, which could however be induced by treatment with SNAP. This suggests that indeed the VMHvl might also be important for mate preferences as suggested by the paper mentioned above. We have changed this accordingly.*

Minor concerns:

54. First two sentences of intro are not informative and could be deleted.

Response: *Changed as requested.*

55. Title sentence for Fig 2 caption needs to be reworded since the data shown addresses the dependence of preference on GnRH and not kisspeptin. The same concern is pertinent to Fig 7 caption.

Response: *Changed as requested.*

56. Title sentence for Fig 6 caption needs reworking. Consider: “Both nNos expression and kisspeptin recipient neurons are increased in the VMHvl in proestrus.”

Response: *The caption for Fig 6 (now Fig 5) has been changed to “VMHvl nNOS neurons are synaptically connected to kisspeptin neurons”.*

57. The data shown in figure 6 do not actually show direct synaptic connections between BL neurons in the VMH and kisspeptin neurons from the AVPV but their data make this likely. I encourage the authors to be more circumspect in how they describe these data.

Response: Changed as requested.

Reviewers' comments:

Reviewer #1 (Remarks to the Author):

This revised manuscript by Hellier and coworkers examines the role of kisspeptin neurons in the RP3V in coordination of sex behaviors and ovulation in females. The authors were responsive to the critiques and the manuscript is much improved.

There are two experiments in which the n is quite low and should be increased. First, the cre negative controls for the optogenetics are only 3 and not consistent so more n are needed. Second, the study in figure 5 has only a n of 4 and rather strong trends; more n should be added to preclude a false negative result.

Page 14 lines 6-7, this statement is best qualified to sex-specific mate preferences and behaviours in females as males were not studied.

Reviewer #2 (Remarks to the Author):

Overall, the authors have done a thorough job of addressing the concerns raised by all three reviewers. Of particular importance, new and more appropriate statistical analyses were conducted. The Results section was reorganized, and additional controls were carried out. The title was modified, and the revised figures and legends are clearer. This reviewer would still encourage the authors to delete the phrase "Rather surprisingly" on p. 10, line 16 in the sentence describing normal levels of lordosis behavior in females with GnRH deficiency. As noted later in the Discussion (p. 14, lines 22-25), it has been known since 1981 that mice failing to make GnRH show normal lordosis behavior when administered appropriate doses of estradiol and progesterone.

To improve readability, the authors may wish to consider breaking up some of the very long paragraphs in the Discussion (especially on pp. 14-17). It can be difficult to follow the main points offered in support of the conclusions. There are also a number of minor punctuation errors, but these should be corrected during copy editing of the final manuscript.

Reviewer #3 (Remarks to the Author):

The revised manuscript is much improved and in general, seems to reflect a genuine effort to thoroughly address the comments and concerns. However, the revised title addresses only one of the two concerns raised without offering an explanation for why it is justified to retain "synchronize female sexual behavior with ovulation." As noted, data for this assertion is simply lacking in the current manuscript. It is not clear that the response to #52 is incorporated into the manuscript.

Responses to reviewers' comments:

Manuscript NCOMMS-17-14543: "A sexually dimorphic neural circuit synchronizes female sexual behavior with ovulation".

We thank the reviewers for their thoughtful and constructive comments on our manuscript. Each of these are addressed as described below, either by changes to the figures, changes to the manuscript, or explanatory responses. All changes in the revised manuscript are highlighted in blue. In addition, we have also made some minor changes to the manuscript in accordance with the formatting requirements of Nature Communications.

Reviewer #1:

General comments

Reviewer #1 (Remarks to the Author):

This revised manuscript by Hellier and coworkers examines the role of kisspeptin neurons in the RP3V in coordination of sex behaviors and ovulation in females. The authors were responsive to the critiques and the manuscript is much improved.

There are two experiments in which the n is quite low and should be increased. First, the cre negative controls for the optogenetics are only 3 and not consistent so more n are needed. Second, the study in figure 5 has only a n of 4 and rather strong trends; more n should be added to preclude a false negative result.

Response: For the optogenetic Cre- controls n was equal to four, however one animal displayed no lordosis in either experiment and therefore was not visible on the original graph. We have now increased the n number to eight and the new results have been incorporated into the manuscript. In regards to figure 5, we have now increased the n number for proestrus to six and metestrus/diestrus to eight and these data have also now been incorporated into the manuscript. As a result, the difference between the number of nNOS+ neurons in the VMHvl is no longer significant and this has been changed accordingly in the manuscript.

Page 14 lines 6-7, this statement is best qualified to sex-specific mate preferences and behaviours in females as males were not studied.

Response: Changed as requested.

Reviewer #2 (Remarks to the Author):

This reviewer would still encourage the authors to delete the phrase "Rather surprisingly" on p. 10, line 16 in the sentence describing normal levels of lordosis behavior in females with GnRH deficiency. As noted later in the Discussion (p. 14, lines 22-25), it has been known since 1981 that mice failing to make GnRH show normal lordosis behavior when administered appropriate doses of estradiol and progesterone.

Response: Deleted as requested.

To improve readability, the authors may wish to consider breaking up some of the very long paragraphs in the Discussion (especially on pp. 14-17). It can be difficult to follow the main points offered in support of the conclusions. There are also a number of minor punctuation errors, but these should be corrected during copy editing of the final manuscript.

Response: We have broken up as the suggested by the reviewer.

Reviewer #3 (Remarks to the Author):

The revised manuscript is much improved and in general, seems to reflect a genuine effort to thoroughly address the comments and concerns. However, the revised title addresses only one of the two concerns raised without offering an explanation for why it is justified to retain "synchronize female sexual behavior with ovulation." As noted, data for this assertion is simply lacking in the current manuscript.

Response: We have now changed the title of the manuscript to “Female sexual behavior is controlled by kisspeptin neurons”.

It is not clear that the response to #52 is incorporated into the manuscript.

Response: This has now been incorporated into the Figure 3 figure legend. “Sham procedures were performed for each intervention as controls. No significant differences between control animals were found; therefore, all controls were combined into a single group.”

REVIEWERS' COMMENTS:

Reviewer #1 (Remarks to the Author):

The authors have responded well to my comments and those of the other reviewers.

Responses to reviewers' comments:

Manuscript NCOMMS-17-14543B: "Female sexual behavior in mice is controlled by kisspeptin neurons".

Reviewer #1:

The authors have responded well to my comments and those of the other reviewers.

Response: We thank the reviewer as well as the other reviewers for their thoughtful and constructive comments on our manuscript.